

**Detection of atmospheric gaseous amines and amides by a high resolution**
**time-of-flight chemical ionization mass spectrometer with protonated**
**ethanol reagent ions**
Lei Yao[1], Ming-Yi Wang[1], Xin-Ke Wang[1], Yi-Jun Liu[1,2], Hang-Fei Chen[1], Jun Zheng[3], Wei Nie[4,5],
Ai-Jun Ding[4,5], Fu-Hai Geng[6], Dong-Fang Wang[7], Jian-Min Chen[1], Douglas R. Worsnop[8], Lin Wang[1,5]*
*[1] Shanghai Key Laboratory of Atmospheric Particle Pollution and Prevention (LAP[3]), Department of*
*Environmental Science & Engineering, Fudan University, Shanghai 200433, China*
*[2] now at Pratt school of engineering, Duke University, Durham, NC 27705,USA*
*[3] Jiangsu Key Laboratory of Atmospheric Environment Monitoring and Pollution Control, Nanjing*
*University of Information Science & Technology, Nanjing 210044, China*
*[4] Joint International Research Laboratory of Atmospheric and Earth System Sciences, School of*
*Atmospheric Science, Nanjing University, 210023, Nanjing, China*
*[5] Collaborative Innovation Center of Climate Change, Nanjing, Jiangsu Province, China*
*[6] Shanghai Meteorology Bureau, Shanghai 200135, China*
*[7] Shanghai Environmental Monitoring Center, Shanghai 200030, China*
*[8] Aerodyne Research, Billerica, MA 01821, USA*
*\* Corresponding Author: L.W., email, lin_wang@fudan.edu.cn; phone, +86-21-65643568; fax,*
*+86-21-65642080.*









**Abstract**
Amines and amides are important atmospheric organic-nitrogen compounds but high time
resolution, highly sensitive, and simultaneous ambient measurements of these species are rather
sparse. Here, we present the development of a high resolution time-of-flight chemical ionization mass
spectrometer (HR-ToF-CIMS) method utilizing protonated ethanol as reagent ions to simultaneously
detect atmospheric gaseous amines ($C_1$ to $C_6$) and amides ($C_1$ to $C_6$). This method possesses
sensitivities of 5.6-19.4 Hz pptv$^{-1}$ for amines and 3.8-38.0 Hz pptv$^{-1}$ for amides under total reagent ion
signals of ~0.32 MHz, and detection limits of 0.10-0.50 pptv for amines and 0.29-1.95 pptv for
amides at 3σ of the background signal for a 1-min integration time, respectively. Controlled
characterization in the laboratory indicates that relative humidity has significant influences on
detection of amines and amides, whereas the presence of organics has no obvious effects. Ambient
measurements of amines and amides utilizing this method were conducted from 25 July 2015 to 25
August 2015 in urban Shanghai, China. While the concentrations of amines ranged from a few pptv to
hundreds of pptv, concentrations of amides varied from tens of pptv to a few ppbv. Among the $C_1$- to
$C_6$-amines, the $C_2$-amines were the dominant species with concentrations up to 130 pptv. For amides,
the $C_3$-amides (up to 8.7 ppb) were the most abundant species. The diurnal profiles of amines and
amides suggest that in addition to the secondary formation of amides in the atmosphere, industrial
emissions could be important sources of amides in urban Shanghai. During the campaign,
photo-oxidation of amines and amides might be a main loss pathway for them in day time, and wet
deposition was also an important sink.



## 1 Introduction

Amines and amides are nitrogen-containing organic compounds widely observed in the
atmosphere (Ge et al., 2011). They are emitted from a variety of natural and anthropogenic sources
including agriculture, biomass burning, animal husbandry, cooking, synthetic leather, carbon capture,
and other industrial processes (Finlayson-Pitts and Pitts, 2000; Ge et al., 2011; Kim et al., 2004; Kuhn
et al., 2011; Nielsen et al., 2012; Zhu et al., 2013). In addition to the primary sources, amides can be
formed from the degradation processes of amines (Nielsen et al., 2012) and atmospheric accretion
reactions of organic acids with amines or ammonia (Barsanti and Pankow, 2006).
Once in the atmosphere, amines and amides can react with atmospheric oxidants (*e.g.*, OH and
$NO_3$ radicals, Cl atoms, and $O_3$), and lead to gaseous degradation products and formation of secondary
organic aerosols (Barnes et al., 2010; Bunkan et al., 2016; El Dib and Chakir, 2007; Lee and Wexler,
2013; Malloy et al., 2009; Murphy et al., 2007; Nielsen et al., 2012). In addition, the basic nature of
amines certainly justifies their participation in atmospheric new particle formation and growth events
(Almeida et al., 2013; Erupe et al., 2011; Glasoe et al., 2015; Smith et al., 2010; Yu et al., 2012;
Zhang et al., 2012). Heterogeneous uptake of amines by acidic aerosols and displacement reactions of
ammonium ions by amines can significantly alter the physio-chemical properties of aerosol particles
(Qiu et al., 2011; Wang et al., 2010a; Wang et al., 2010b).
Atmospheric amines have been measured in different surroundings. Kieloaho et al. (2013) used
offline acid-impregnated fiberglass filter collection together with analysis by high performance liquid
chromatography electrospray ionization ion trap mass spectrometer, and reported that the highest
concentrations of $C_2$-amines (ethylamine (EA) + dimethylamine (DMA)) and $C_3$-amines (propylamine
(PA) + trimethylamine (TMA)) reached 157±20 pptv (parts per trillion by volume) and 102±61 pptv,
respectively, in boreal forests, southern Finland. Using a similar detection method, the mean
concentrations of $C_2$-amines (EA+DMA), $C_3$-amines (PA+TMA), butylamine (BA), diethylamine
(DEA) and triethylamine (TMA) were measured to be 23.6 pptv, 8.4 pptv, 0.3 pptv, 0.3 pptv, and 0.1
pptv, respectively, in urban air of Helsinki, Finland (Hellén et al., 2014). Detection of gaseous alkyl
amines were conducted in Toronto, Canada using an ambient ion monitor ion chromatography system,
and the concentrations of DMA, and TMA+DEA were both less than 2.7 ppt (VandenBoer et al. 2011).
Recently, online detection of atmospheric amines using chemical ionization mass spectrometer is
becoming the trend. Yu and Lee (2012) utilized a quadrupole chemical ionization mass spectrometer





(CIMS) with protonated ethanol and acetone ions as reagent ions to measure $C_2$-amines ($8\pm3$ pptv)
and $C_3$-amines ($16\pm7$ pptv) in Kent, Ohio. A similar method detected a few pptv to tens of pptv
$C_3$-amines in Alabama forest (You et al., 2014). Sellegri et al. (2005) reported the mean concentration
of TMA and DMA were 59 pptv and 12.2 pptv, respectively, in Hyytiälä forest, with a
quadrupole-CIMS with hydronium ions as reagent ions. Measurements of amines in urban areas did
not show significant differences in terms of concentration. The average of total amines ($C_1$-$C_3$) was
$7.2\pm7.4$ pptv in Nanjing, China, as measured by a high resolution time-of-flight CIMS
(HR-ToF-CIMS) with hydronium ions as reagent ions (Zheng et al., 2015). Measurements by an
ambient pressure proton transfer mass spectrometer (AmPMS) in urban Atlanta showed that
trimethylamine (TMA) (or isomers or amide) was the most abundant amine-species and that the
concentration of DMA was ~3 pptv (Hanson et al., 2011).
To the best of our knowledge, gaseous amides were not previously measured in ambient air,
except for two studies that briefly described the detection of a few amides near the emission source.
Zhu et al. (2013) detected formamide (FA) ($C_1$-amide) formed from degradation of
mono-ethanolamine in emissions from an industrial carbon capture facility, using proton transfer
reaction time-of-flight mass spectrometry (PTR-ToF-MS). Furthermore, up to 4350 pptv of
dimethylamide was observed near a municipal incinerator, waste collection center and sewage
treatment plant (Leach et al., 1999).
Given the important role of amines in atmospheric nucleation and other physio-chemical
processes, and the potential involvement of amides in a number of atmospheric processes, it is
necessary to develop high time resolution and highly sensitive detection techniques for measurements
of ambient amines and amides. Previous studies have proven CIMS to be a powerful instrument to
detect gaseous amines and amides in laboratory studies and field campaigns (Borduas et al., 2015;
Bunkan et al., 2016; Hanson et al., 2011; Sellegri et al., 2005; Yu and Lee, 2012; You et al., 2014;
Zheng et al., 2015). However, the detection method for ambient amides with much lower
concentrations than those in laboratory studies is still lacking. The popular usage of hydronium ions
as reagent ions (*e.g.* PTR-MS and AmPMS) potentially leads to the relative humidity (RH)
dependence of the background and ambient amine signals, adding uncertainties to measurement
results (Hanson et al., 2011; Zheng et al., 2015; Zhu et al., 2013). In addition, constrained by the mass
resolution of the quadrupole-detector mass spectrometer, it is difficult to distinguish protonated



amines and amides with an identical unit mass, which pre-excludes the possibility of simultaneous
measurements of amines and amides. For example, the *m/z* (mass to charge ratio) value of protonated
trimethylamine ($C_3H_9N \cdot H^+$, *m/z* 60.0808) and that of protonated acetamide ($C_2H_5NO \cdot H^+$, *m/z* 60.0444)
are very close.

In the present study, a HR-ToF-CIMS method utilizing protonated ethanol as reagent ions has

been developed to simultaneously detect atmospheric gaseous amines ($C_1$ to $C_6$) and amides ($C_1$ to $C_6$).
Protonated ethanol was selected as the reagent ions because the higher proton affinity of ethanol
(185.6 kcal mol$^{-1}$) than that of water (165.2 kcal mol$^{-1}$), as shown in Table 1, results in more
selectivity for detecting high proton affinity species (*e.g.* > 196 kcal mol$^{-1}$ for amines and amides)
(Nowak, 2002; Yu and Lee, 2012; You et al., 2014). The influences of RH and organics on amine and
amide detection were characterized in the laboratory. Ambient measurements of amines and amides
utilizing this method were performed from 25 July 2015 to 25 August 2015 in urban Shanghai, China.
During the campaign, a Filter Inlet for Gases and AEROsols (FIGAERO) was interfaced to
HR-ToF-CIMS (Lopez-Hilfiker et al., 2014) but only results on gaseous $C_1$-$C_6$ amines and amides are
presented. The potential sources and sinks of amines and amides are discussed.

**2 Experimental**
**2.1 Instrumentation**

An Aerodyne HR-ToF-CIMS (Bertram et al., 2011) with protonated ethanol as reagent ions has

been deployed to detect gaseous amines ($C_1$ to $C_6$) and amides ($C_1$ to $C_6$). Protonated ethanol reagent
ions were generated by passing a pure air flow of 1 L min$^{-1}$ (liter per minute) supplied by a zero air
generator (Aadco 737) through a Pyrex bubbler containing ethanol (≥ 96%, J.T.Baker) and then
through a 0.1 mCi $^{241}$Am radioactive source. A sample flow of 1.35 L min$^{-1}$ was introduced into the
ion-molecular reaction (IMR) chamber where the sample flow and the reagent ion flow converge. The
pressures of the IMR chamber and the small-segmented quadrupole (SSQ) were regulated at ~100
mbar and ~2.8 mbar, respectively, to increase the instrument sensitivity. Under such conditions, the
ion-molecular reaction time was ~320 ms in the IMR. To minimize wall-loss of analytes on the inner
surface of IMR, the temperature of IMR was maintained at an elevated temperature (50°C). The data
of HR-ToF-CIMS was collected at 1 Hz time resolution.

Under dry conditions, the most abundant reagent ion was protonated ethanol dimer





$((C_2H_5OH)_2 \cdot H^+$, $m/z$ 93.0910), with the next dominant ions being protonated ethanol monomer
( $(C_2H_5OH) \cdot H^+$, $m/z$ 47.0491) and protonated ethanol trimer $((C_2H_5OH)_3 \cdot H^+$, $m/z$ 139.1329). The
presence of water led to formation of clusters of protonated ethanol with water $(C_2H_5OH \cdot H_2O \cdot H^+)$ and
hydronium ions and their hydrates $((H_2O)_n \cdot H^+$, n=1, 2 and 3). A typical mass spectrum under < 20%
RH is shown in Figure S1. The ratios of oxygen cation $(O_2^+)$, the clusters of protonated ethanol with
water $(C_2H_5OH \cdot H_2O \cdot H^+)$, and hydronium ions $((H_2O)_n \cdot H^+$, n=1, 2 and 3) to the sum of $(C_2H_5OH) \cdot H^+$,
$(C_2H_5OH)_2 \cdot H^+$, and $(C_2H_5OH)_3 \cdot H^+$ were ~0.001, ~0.026, and ~0.011, respectively.
Amines and amides reacted dominantly with protonated ethanol ions through proton transfer
reactions. The ion-molecular reactions of amines (denoted as $NR_3$, R=H or an alkyl group) and amides
(denoted as $R'_2NC(O)R$, R'=H or an alkyl group) with protonated ethanol reagent ions can be showed
as the following (You et al., 2014; Yu and Lee, 2012):
$(C_2H_5OH)_n \cdot H^+ \ + \ NR_3 \ \longrightarrow \ (C_2H_5OH)_j \cdot NR_3 \cdot H^+ + (n\text{-}j)C_2H_5OH$            (R1)
$(C_2H_5OH)_n \cdot H^+ + R'_2NC(O)R \longrightarrow \ (C_2H_5OH)_j \cdot R'_2NC(O)R \cdot H^+ +(n\text{-}j)C_2H_5OH$       (R2)
where $n$ = 1, 2, and 3, and $j$ = 0 and 1.
**2.2 Calibration of amines and amides**
Amines and amides were calibrated using permeation sources. Permeation tubes for amines
(methylamine (MA), dimethylamine (DMA), trimethylamine (TMA), and diethylamine (DEA)) were
purchased from VICI Inc. USA, whereas those for amides (formamide (FA), $\geqslant$ 99.5%, GC, Sigma
Aldrich; acetamide (AA), $\geqslant$ 99%, GC, Sigma Aldrich; and propanamide (PA), $\geqslant$ 96.5%, GC, Sigma
Aldrich) were home-made using 1/4 inch Teflon tubes with the ends sealed with Teflon blockers. The
permeation tube was placed in a U-shape glass tube with a diameter of 2.5 cm that was immersed in a
liquid bath with precise temperature regulation (Zheng et al., 2015). A high purity ($\geqslant$99.999%)
nitrogen flow typically at 0.1 L min$^{-1}$ was used as the carrier gas to carry the permeated compounds to
HR-ToF-CIMS for detection.
The concentration of an amine in the outflow of the permeation tube was referred to that as
determined by an acid-base titration method (Freshour et al., 2014). The high purity nitrogen flow
containing an amine standard was bubbled through a HCl solution (pH = ~4.5) with a small amount of
KCl (~5 mM) added to facilitate measurements of pH values. Reagents HCl (~37 wt% in water) and
KCl ($\geqslant$ 99%) were of ACS reagents and purchased from Sigma Aldrich. The concentration of the



amine was derived according to variations of the pH values with titration time. The pH values were
averaged and recorded by a pH meter (340i, WTW, Germany) every 5 min.
In the case of amides, a permeated alkyl-amide was trapped in a $HNO_3$ solution with a pH of
~2.5 that was diluted from reagent $HNO_3$ (~70 wt% in water, ACS reagent, Sigma Aldrich).
Hydrolysis of the alkyl-amide occurred under acidic conditions leading to formation of $NH_4^+$ (Cox and
Yates, 1981). The concentration of $NH_4^+$ was quantified using Ion Chromatography (Metrohm 833,
Switzerland), and the permeation rate of the alkyl-amide was derived from the variation of $NH_4^+$ with
the time period of hydrolysis.
The total ethanol reagent ion signals during the laboratory calibration were typically ~0.32 MHz.
**2.3 Influence of RH and organics**
Experiments were performed to characterize the influence of RH and organics on the detection of
amines and amides. The schematics of our experimental setup are shown in Figure S2 (A for tests at
elevated RH and B for tests in presence of organics), where the tubes and valves are made of
polytetrafluoroethylene (PTFE) and perfluoroalkoxy (PFA) materials to minimize absorption of
amines/amides on the inner surface of tubes and valves. To examine the influence of RH, a pure air
flow was directed through a bubbler filled with 18.2 MΩ·cm deionized water, and then mixed with the
amine/amide flow of 0.1 L min$^{-1}$ generated from the permeation sources. The examined RH ranged
from 4 % to 65%.
α-pinene, a typical biogenic organic compound, and p-xylene, a typical anthropogenic compound,
were chosen to examine the influence of organics on detection of amines and amides. The
amines/amide flow was mixed with organics for ~0.2 s before entering the IMR. During the
characterization, the air flow (15 L min$^{-1}$) containing α-pinene or p-xylene with concentrations up to
hundreds of ppbv (parts per billion by volume) was initially mixed the amine/amide flow of 0.1 L
min$^{-1}$ generated from the permeation sources. Then ozone and OH radicals were generated from an
$O_2/H_2O$ flow of $2 \times 10^{-3}$ L min$^{-1}$ by turning on a Hg-lamp (Jelight model 600, USA). Photochemical
reactions of α-pinene or p-xylene occurred and a much more complex mixture of organics was
subsequently mixed with the amine/amide flow.
**2.4 Field campaign in urban Shanghai**
The ethanol HR-ToF-CIMS was deployed for a field campaign at the Fudan site (31° 17′ 54″ N,





121° 30′ 05″ E) on the campus of Fudan University from 25 July through 25 August, 2015. This
monitoring site is on the rooftop of a teaching building that is ~20 m above ground. About 100 m to
the north is the Middle Ring that is one of the main overhead highways in Shanghai. This site is also
influenced by local industrial and residential activities. Hence, the Fudan site is a representative urban
site (Ma et al., 2014; Wang et al., 2013a; Wang et al., 2016; Xiao et al., 2015).

The schematic of the ethanol HR-ToF-CIMS setup during the field campaign is shown in Figure

S3. Ambient air was drawn into a PTFE tubing with a length of 2 m and an inner diameter of 3/8 inch.
To minimize the wall-loss of amines and amides, a high sampling flow rate (15 L min$^{-1}$) was adopted,
resulting in an inlet residence time of ~0.26 s. Also, the PTFE tubing was heated to 50℃ by heating
tapes. Because of the high concentrations of volatile organic compounds in the air of urban Shanghai,
reagent ion depletion occurred during the initial tests of measurements of ambient samples. Hence, the
ambient air was diluted with a high purity nitrogen flow with a dilution ratio of ~1:4.6. Under such
condition, variation of the total reagent ions (($C_2H_5OH$)·$H^+$, ($C_2H_5OH$)$_2$·$H^+$, and ($C_2H_5OH$)$_3$·$H^+$) was
less than 10% between measurements of the background air and the ambient sample. The ethanol
reagent ion signals were typically around 0.35 MHz throughout the entire campaign. To take the
variation in total reagent ions between in laboratory calibration and during field measurements into
account, ambient concentrations of amines and amides were calculated according to
$$[\text{amines or amides}]_{\text{ambient}} = C_{\text{amines or amides}} \times \frac{\sum_{n=0-1}(\text{amines or amides})\cdot(C_2H_5OH)_n\cdot H^+}{\sum_{n=1-3}(C_2H_5OH)_n H^+} \quad (1)$$
where C is a calibration coefficient obtained by dividing the total reagent ion signals in laboratory
calibration by the sensitivity of an amine or amide. As shown in equation (1), to minimize the effect of
the variation of reagent ions during field measurements, the ambient signals of an amine or amide
were normalized by the sum of ethanol clusters including protonated ethanol monomer, dimer, and
trimer.

During the campaign, a Filter Inlet for Gases and AEROsols (FIGAERO) (Lopez-Hilfiker et al.,

2014) was attached to the HR-ToF-CIMS. FIGAERO-HRToF-CIMS offers two operation modes.
Direct gas sample analysis occurs with the HR-ToF-CIMS during simultaneous particle collection on
a PTFE filter via a separate dedicated port. Particle analysis occurs via evaporation from the filter
using temperature-programmed thermal desorption by heated ultra-high purity nitrogen upstream of
the HR-ToF-CIMS. A moveable filter housing automatically switches between the two modes. In our





study, measurements of ambient gaseous samples were conducted for 20 min every hour, followed by analysis of particulate samples for 40 min. In this paper, we focus on measurements of gaseous samples and present results on detection of gaseous amines and amides.

During the 20 min period for analysis of ambient gaseous samples, background measurements were auto-performed for 5 min by a motor-driven three-way Teflon solenoid valve, utilizing a high purity nitrogen flow as the background gas. Figure S4 shows typical background signals during an ambient sampling period of 3 h. The average ambient background concentrations of amines ($C_1$ to $C_6$) and amides ($C_1$ to $C_6$) throughout the field campaign are presented in Table 1. The inlet memory of amines and amides were determined using an inlet spike approach. As shown in Figure S5, the signals followed the sum of two decaying exponentials. The characteristic decaying times of two exponentials, which are displacement of amines and amides inside the inlet by pumping and removing amines and amides adsorbed on the inlet surface (Zheng et al., 2015), were 1.1 s and 8.5 s for TMA, and 1.4 s and 1.4 s for PA, respectively. These results demonstrate that a 5 min background sampling time is sufficient to eliminate the inlet memory.

All HR-ToF-CIMS data were analyzed with Tofware (Aerodyne Research, Inc. and Tofwerk AG) and Igor Pro (Wavemetrics) software. Concentrations of $O_3$ were measured by an $O_3$ analyzer (Model 49i, Thermo Scientific, USA). RH and temperature were measured by an automatic meteorological station (CAWS600, Huayun, China) at the Fudan site.

Solar radiation intensity measured by a pyranometer (Kipp & Zonen CMP6, Netherlands) was obtained from the Shanghai Pudong Environmental Monitoring Centre (31°14′ N, 121°32′ E, about 8.78 km from the Fudan site). Precipitation was recorded by a rainfall sensor (RainWise Inc., USA) located at the Huangxing Park monitoring station (31°17′ N, 121°32′ E, about 2.95 km from the Fudan site) of Shanghai Meteorology Bureau.

## 3 Results and Discussion

### 3.1 Performance of ethanol HR-ToF-CIMS in the laboratory

#### 3.1.1 Sensitivities and detection limits

The permeation rates of amines and amides were determined adopting methods of acid-base titration and hydrolysis of alkyl-amides in an acidic solution, respectively. A typical plot for





determination of the permeation rate of the DEA permeation tube is shown in Figure S6. Plots for FA
($C_1$-amide) and PA (an isomer of $C_3$-amide) are used as examples for amides, as shown in Figure S7.
In summary, at 0℃, the permeation rates of MA, DMA, TMA, and DEA permeation tubes were 6.9 ±
0.7, 7.4 ± 0.2, 5.1 ± 0.8, and 12.7 ± 0.9 ng min$^{-1}$, respectively. Permeation rates of home-made FA,
AA, and PA permeation tubes were 36.7 ± 2.4, 5.2 ± 0.5, and 29.1 ± 1.6 ng min$^{-1}$, respectively, at 0℃.
The high purity nitrogen flow carrying the permeated amine/amide was then diluted with another
high purity nitrogen flow at different dilution ratios, and directed to HR-ToF-CIMS for detection
under dry conditions (RH = ~ 0%). Figure S8 shows the calibration curves of $C_1$- to $C_4$-amines and
$C_1$- to $C_3$-amides. The derived sensitivities were 5.6-19.4 Hz pptv$^{-1}$ for amines and 3.8-38.0 Hz pptv$^{-1}$
for amides with the total reagent ions of ~0.32 MHz, respectively. Also, the detection limits of amines
and amides were 0.10-0.50 pptv and 0.29-1.95 pptv at 3σ of the background signal for a 1-min
integration time, respectively. Sensitivities, calibration coefficients, and detection limits of the $C_1$- to
$C_4$-amines (MA, DMA, TMA, and DEA) and $C_1$- to $C_3$-amides (FA, AA, and PA), together with their
proton affinities, are summarized in Table 1. The detection limits of $C_1$- to $C_3$-amines in our study are
similar to those by Zheng et al. (2015) and You et al. (2014). The sensitivities of $C_1$- to $C_4$-amines are
slightly better than those reported in You et al. (2014) and Yu and Lee (2012).

**3.1.2 Effects of RH and organics**

The presence of high concentrations of water is believed to have an effect on the ion-molecular
reactions in IMR, given the proton transfer nature of our ion-molecular reactions and the high IMR
pressure (providing longer ion-molecular reaction time) in our study. The detection of constant
concentrations of amines and amides by HR-ToF-CIMS at various RH was characterized to evaluate
the influence of RH. Examined were MA ($C_1$-amine) and TMA ($C_3$-amine) under 0-65% RH at 23℃,
corresponding to 0-70% and 0-49% enhancement in the MS signal, respectively. In the case of amides,
the increase of the PA ($C_3$-amide) signal was 0-38% under 0-55% RH. These results show that RH has
an obvious effect on the MS signals for amines and amides, which followed sigmoidal fits with the
$R^2 \geqslant 0.97$ in the examined RH range (Figure 1).
Figure 2 shows the effects of biogenic (α-pinene) and anthropogenic (p-xylene) compounds and
their photochemical reaction products on detection of amines (MA and TMA) and amide (PA) by our
HR-ToF-CIMS. After stable signals of amines/amide were established, introduction of α-pinene and





p-xylene, respectively, had little impact on detection of amines and amides. Initiation of
photochemical reactions of α-pinene and p-xylene upon turning on the Hg-lamp, as evidenced by
characteristic products of pinonaldehyde from α-pinene (Lee et al., 2006) and 3-hexene-2,5-dione
from p-xylene (Smith et al., 1999), respectively, did not have an obvious effect on detection of amines
and amides, either.

### 3.2 Detection of amines and amides in urban Shanghai

### 3.2.1 Identification of nitrogen-containing species

One major challenge during analysis of mass spectra from the field deployment of the ethanol
HR-ToF-CIMS is to distinguish amines and amides with very close $m/z$ values in order to achieve
simultaneous measurements. Thanks to the high mass resolving power ($R \geq 3500$ in V-mode) of our
HR-ToF-CIMS, we are able to distinguish and identify the following protonated amines ($CH_5N \cdot H^+$
($m/z$ 32.0495), $C_2H_7N \cdot H^+$ ($m/z$ 46.0651), $C_3H_9N \cdot H^+$ ($m/z$ 60.0808), $C_4H_{11}N \cdot H^+$ ($m/z$ 74.0964),
$C_5H_{13}N \cdot H^+$ ($m/z$ 88.1121), and $C_6H_{15}N \cdot H^+$ ($m/z$ 102.1277)), and amides ($CH_3NO \cdot H^+$ ($m/z$ 46.0287),
$C_2H_5NO \cdot H^+$ ($m/z$ 60.0444), $C_3H_7NO \cdot H^+$ ($m/z$ 74.0600), $C_4H_9NO \cdot H^+$ ($m/z$ 88.0757), $C_5H_{11}NO \cdot H^+$ ($m/z$
102.0913), and $C_6H_{13}NO \cdot H^+$ ($m/z$ 116.1069)), as well as a few oxomides ($C_3H_5NO_2 \cdot H^+$ ($m/z$ 88.0393),
$C_4H_7NO_2 \cdot H^+$ ($m/z$ 102.0550), and $C_5H_7NO_2 \cdot H^+$ ($m/z$ 116.0760)), as shown by the single peak fitting
for each of them in Figure 3. The assignment of molecular formulas for these species is within a mass
tolerance of < 10 ppm.
We further analyzed the entire mass spectra and assigned a molecular formula to 200 species
with $m/z$ values less than 163 Th as listed in Table S1, which allows a mass defect plot for typical
15-min mass spectra in Figure 4A. In addition to the protonated $C_1$ to $C_6$-amines and amides, their
clusters with ethanol are evident, further confirming the identification of these species. A number of
gaseous amines have been previously detected in the ambient air utilizing quadrupole mass
spectrometer (Freshour et al., 2014; Hanson et al., 2011; Sellegri et al., 2005; You et al., 2014; Yu and
Lee, 2012). As suggested by Hanson et al. (2011), an amine and an amide with one less carbon might
both have high enough proton affinities and could be detected at the same unit $m/z$ value by a
quadrupole mass spectrometer, leading to uncertainty in measuring the ambient amine. In this study,
$C_1$- to $C_6$-amines and $C_1$- to $C_6$-amides are, for the first time, systematically and simultaneously
detected in ambient air.





In addition to the protonated $C_1$ to $C_6$-amines and $C_1$ to $C_6$-amides and their clusters with ethanol,
we were able to detect many other nitrogen-containing species (*e.g.* ammonia). Among the 200
species with *m/z* less than 163 Th, there were 86 nitrogen-containing species (Figure 4B and Table S1).
Four imines (or enamines) including $CH_3N$, $C_2H_5N$, $C_3H_7N$, and $C_4H_9N$ were detected. These imines
(or enamines) could derive from photo-oxidation of amines (Nielsen et al., 2012). In addition, a
number of heterocyclic nitrogen-containing species including pyrrole, pyrroline, pyrrolidine, pyridine,
and pyrimidine were potentially detected (see Table S1). Apart from clusters of ammonia, $C_1$- to
$C_6$-amines, and $C_1$- to $C_6$-amides with water or ethanol, there were forty-eight $C_aH_bN_cO_d$ species
representing 55.8 % of the total nitrogen-containing species. This suggests that more than half of the
nitrogen-containing species existed as oxygenated compounds in the atmosphere in urban Shanghai.
The rest 114 species with *m/z* less than 163 Th are mostly organics (see Table S1). Above
*m/z*=163 Th, numerous mass peaks were observed, which are likely organics and nitrogen-containing
species. These high-molecular-weight species are assumed to have a low volatility and may partition
between the gas phase and the particles.
**3.2.2 Time profiles of amines and amides**
During the field measurement, the average RH of the diluted gaseous samples was 15.8±3.5%.
According to our laboratory characterization, the MS signals of MA, TMA, and PA at 15.8% RH have
been in average enhanced by 10%, 9%, and 19%, respectively, from our calibration under dry
condition. Here, we use our sigmoidal fits to convert each of our ambient data points to the signal
under dry condition (RH = ~0%), and calculate the corresponding concentration. Since MA and TMA
behaved quite similarly at elevated RH, the sigmoidal fit for TMA is also applied to the $C_2$-amines and
$C_4$- to $C_6$-amines. Also, the sigmoidal fit for PA is adopted for other amides. Since high purity
nitrogen (RH = ~0%) was used as the background sample during the ambient campaign, no
RH-dependent correction was made with background signals.
Assuming $C_1$- to $C_4$-amines have the same proton affinity as MA, DMA, TMA, and DEA,
respectively, the sensitivities of MA, DMA, TMA, and DEA were used to quantify $C_1$- to $C_4$-amines.
Since $C_5$- to $C_6$-amine standards are unavailable, the sensitivity of DEA by HR-ToF-CIMS was
adopted to quantify $C_5$- to $C_6$-amines. A similar approach was utilized to quantify $C_1$- to $C_3$-amides by
sensitivities of FA, AA, and PA, respectively. In addition, the sensitivity of PA was used to quantify





$C_4$- to $C_6$-amides.

Figure 5 presents the time profiles for mixing ratios of $C_1$- to $C_6$-amines and $C_1$- to $C_6$-amides,

respectively, from 25 July to 25 August 2015 in urban Shanghai. Note that each date point in the
figure represents an average of 15-min measurements. Table 2 summarizes the mean concentrations of
$C_1$- to $C_6$-amines and $C_1$- to $C_6$-amides throughout the entire campaign, together with comparison of
amine and amide concentrations reported in previous field studies.

For $C_1$- to $C_6$-amines, the average concentrations ($\pm\sigma$) were 15.7±5.9 pptv, 40.0±14.3 pptv,

1.1±0.6 pptv, 15.4±7.9 pptv, 3.4±3.7 pptv, and 3.5±2.2 pptv, respectively. $C_1$-amine, $C_2$-amines, and
$C_4$-amines were the dominant amine species in urban Shanghai. The concentrations of amines in
Shanghai are generally smaller than those in Hyytiälä, Finland (Hellén et al., 2014; Kieloaho et al.,
2013; Sellegri et al., 2005), potentially hinting that sources for amines existed in the forest region of
Hyytiälä, Finland. Our $C_1$- and $C_2$-amines are generally more abundant than those in coastal,
continental, suburban, and urban areas (Freshour et al., 2014; Hanson et al., 2011; Kieloaho et al.,
2013; Sellegri et al., 2005; You et al., 2014). However, our $C_3$- to $C_6$-amines are less, potentially
because we are able to distinguish an amine, an amide with one less carbon, and an oxoamide with
two less carbons (see Figure 3).

For $C_1$- to $C_6$-amides, the average concentrations ($\pm\sigma$) were 2.3±0.7 pptv, 169.2±51.5 pptv,

778.2±899.8 pptv, 167.8±97.0 pptv, 34.5±13.3 pptv, and 13.8±5.2 pptv, respectively. $C_2$-amides,
$C_3$-amides, and $C_4$-amides were the most abundant amides in urban Shanghai during the campaign
and their concentrations were up to hundreds of pptv. Especially, the concentration of $C_3$-amides
reached ~8700 pptv. Up to now, studies that report systematical identification and quantification of
amides in the ambient air are lacking. Leach et al. (1999) detected *N,N*-dimethylformamide (an isomer
of $C_3$-amides) of 368-4357 pptv in a suburban area surrounded by municipal incinerator, waste
collection and processing center, and sewage treatment plant. In the ambient air, $C_1$- to $C_6$-amides may
derive from oxidation of $C_1$- to $C_6$-amines. *N,N*-dimethylformamide is a major product with a yield of
~40% from photolysis experiments of TMA under high $NO_x$ conditions (Nielsen et al, 2011). Also, the
yields of formamide ($C_1$-amide) and methylforamide ($C_2$-amide) from OH-initiated MA and DMA in
the presence of $NO_x$ are ~11% and ~13%, respectively (Nielsen et al, 2012). Comparison of the
abundance of amines and amides during the campaign, together with the yields of amides from





photo-oxidation of amines, suggests that the ambient $C_1$- to $C_3$-amines were insufficient to explain the
observed abundance of $C_1$- to $C_3$-amides. Therefore, in addition to secondary sources, $C_1$- to
$C_6$-amides likely were emitted from primary sources (*e.g.* industrial emissions).
Figure 6 shows a close examination on the temporal variations of $C_2$-amines and $C_3$-amides,
representatives of the observed amines and amides, together with that of rainfall between 20 August
2015 and 25 August 2015. The plots clearly reveal that the concentrations of $C_2$-amines and
$C_3$-amides on raining days were much lower than those without rain, and that $C_2$-amines and
$C_3$-amides rapidly went up after the rain. Previous studies reported that wet deposition is one of the
important sinks of amines (Cornell et al., 2003; Ge et al., 2011a, b; You et al., 2014). Our study further
indicates that wet deposition (or heterogeneous reactions) is also an important sink for amides.

### 405    3.2.3 Diurnal patterns

Figure 7 presents the averaged diurnal variations of $C_1$- and $C_2$-amines and $C_3$- and $C_4$-amides,
together with those of temperature, radiation, and ozone concentration during the campaign. Diurnal
patterns for amines and amides with less variation are exhibited in Figure S9. Mixing ratios of $C_1$- and
$C_2$-amines and $C_3$- and $C_4$-amides reached their peak values in the early morning (6:00~7:00am), and
then started to decline as the temperature increased. The mixing ratios were normally the lowest
during the day when the temperature rose to the top. The diurnal behavior of amines and amides can
be explained by the strong photochemical reactions of these species during the daytime (Barnes et al.,
2010; Borduas et al., 2015; Nielsen et al., 2012), especially in summer, as evidenced by the negative
correlations between the mixing ratios and radiation (exponential fits with -0.0002≤ exponents ≤
-0.0001), and between the mixing ratios and ozone (exponential fits with -0.003≤ exponents ≤
-0.001), a tracer for photochemical activities. Also, nighttime chemistry of amines with $NO_3$ radicals
could be active. In summer night time of Shanghai, the $NO_3$ radical concentration could be up to $10^{10}$
radicals $cm^{-3}$ (Wang et al., 2013b) and the reaction rates of amines with $NO_3$ radicals are at the order
of $10^{-13}$ $cm^3$ $molecular^{-1}s^{-1}$ (Nielsen et al., 2012). Hence, high mixing ratios of amines at nighttime
could be a secondary source of amides through reactions of amines with $NO_3$ radicals.
In addition, an opposite tendency between the mixing ratios and the temperature (exponential fits
with -0.067≤ exponents ≤-0.049) is clearly evident in our study, which is in contrast to the positive
temperature dependence of $C_3$-amines and $C_6$-amines in previous studies (Hanson et al., 2011; You et





al., 2014; Yu and Lee, 2012). The positive temperature dependencies of $C_3$-amines was explained by
deposition of amines onto soil or grass landscape at night and then partitioning back to the atmosphere
in the morning when the surface heats (Hanson et al., 2011; You et al., 2014). On the other hand, land
surface in Shanghai is mainly covered by bitumen and cement, on which the behavior of amines might
be different.
**3.2.4 Source identification for $C_3$-amides**
Lagrangian dispersion model has been utilized to further understand the potential sources of
$C_3$-amides. This Lagrangian modeling simulation is based on Hybrid Single-Particle Lagrangian
Integrated Trajectory (HYSPLIT) (Draxler and Hess, 1998; Stein et al., 2015) following the method
developed by Ding et al. (2013). Three-day backward retroplumes (100 m above the ground level)
from the Fudan sampling site are demonstrated for air masses with mixing ratios of $C_3$-amides > 2670
pptv in Figure 8, and for air masses with 2650 pptv > mixing ratios of $C_3$-amides > 1340 pptv in
Figure S10, respectively. The embedded 12 h retroplumes give a better view of the local zones
through which the air masses with high concentrations of $C_3$-amides passed before their arrival to the
sampling site. Since the atmospheric lifetimes of *N,N*-dimethylformamide (an isomer of $C_3$-amides)
and its potential precursor TMA (an isomer of $C_3$-amines) in respect to reactions with OH radicals are
~3 h and ~10 h, respectively, using an 12 h-average OH radical concentration of $2 \times 10^6$ radicals $cm^{-3}$,
$C_3$-amides were likely emitted or formed along the trajectory. As shown in Figure 8A-D, the air
plumes with high concentrations of $C_3$-amides mainly originated from the sea and came from the
north of Shanghai. The air mass passed through predominantly industrial areas and cities after landing,
and Baoshan industrial zone (one of the main industrial zones in Shanghai) was right on its path during
the last 12 hours. Therefore, industrial emissions (or other anthropogenic emissions) might be
important sources of $C_3$-amides.
Figure S10A-E presents another five cases with the next highest concentrations of $C_3$-amide. The
air masses primarily came from southwest of the sampling site, and then passed through industrial
areas and cities before arrival, including Songjiang and Jinshan industrial zones (another two main
industrial zones in Shanghai) during the last 12 hours. These results also suggest that industrial
emissions or other anthropogenic activities might be important sources of $C_3$-amides.



## 4 Conclusions


This paper presents laboratory characterization of an ethanol HR-ToF-CIMS method for
detection of amines and amides, and one month field deployment of the ethanol HR-ToF-CIMS in
urban Shanghai during summer 2015. Laboratory characterization indicates that our sensitivities for
amines (5.6-19.4 Hz pptv$^{-1}$) and amides (3.8-38.0 Hz pptv$^{-1}$) and detection limits for amines
(0.10-0.50 pptv) and amides (0.29-1.95 pptv at $3\sigma$ of the background signal for a 1-min integration
time), respectively, are slightly better than those in previous studies using CIMS methods (You et al.,
2014; Yu and Lee, 2012). Correction of the mass signals of amines and amides are necessary at
elevated RH because of the significant RH dependence of detection of amines and amides as observed
in the laboratory. On the other hand, organics with high proton affinity are unlikely to pose an effect
on the detection of amines and amides as along as their concentrations will not lead to reagent ion
depletion.
High time resolution, highly sensitive and simultaneous measurements of amines (from a few
pptv to hundreds of pptv) and amides (from tens of pptv to a few ppbv) have been achieved during the
ambient campaign. Their diurnal profiles suggest that primary emissions could be important sources
of amides in urban Shanghai, in addition to the secondary formation processes, and that
photo-oxidation and wet deposition of amines and amides might be their main loss pathway.
86 nitrogen-containing species including amines and amides were identified with *m/z* less than
163 Th, 55.8% of which are oxygenated. This certainly indicates that the ethanol HR-ToF-CIMS
method potentially has a much wider implication in terms of measuring atmospheric
nitrogen-containing species. For example, imines (or enamines) and a number of heterocyclic
nitrogen-containing compounds (*e.g.* pyridine and quinoline) (see Table S1) were potentially detected
by this method. Berndt et al. (2014) reported that pyridine was able to enhance nucleation in
$H_2SO_4/H_2O$ system. Also, proton affinities of most of these heterocyclic nitrogen-containing
compounds are higher than that of ammonia, hence they potentially have the capacity to neutralize
atmospheric acidic species (*e.g.* $H_2SO_4$, $HNO_3$ and organic acids) to contribute to secondary particle
formation and growth.
Nevertheless, the detection of amides in ambient air is consistent with the photochemical
chemistry that has been previously studied in the laboratory (Nielsen et al, 2012). Compared with
amines, acetamide has a very weak positive enhancement on nucleation capability of sulfuric acid



(Glasoe et al., 2015). On the other hand, the mixing ratios of amides were significantly higher than
those of amines in urban Shanghai during our measurements. Since the newly formed nano-particles
are likely highly acidic (Wang et al., 2010a), hydrolysis of amides will give rise to $NH_4^+$ in the particle,
in addition to those formed through direct neutralization between gaseous ammonia and particulate
sulfuric acid. Although significant progresses on the roles of ammonia and amines in the atmospheric
nucleation have been made (Almeida et al., 2013; Kürten et al., 2014) and it has been shown that
acetamide can only slightly enhance the nucleation rate of sulfuric acid (Glasoe et al., 2015), the exact
contribution of amides during atmospheric nucleation and subsequent growth events is yet to be
elucidated.

**Acknowledgement**
This study was financially supported by the National Natural Science Foundation of China (No.
21190053, 21222703, 21561130150, & 41575113), the Ministry of Science & Technology of China
(2012YQ220113-4), and the Cyrus Tang Foundation (No. CTF-FD2014001). LW thanks the Royal
Society-Newton Advanced Fellowship (NA140106).



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





Table 1. Proton affinity, sensitivity, calibration coefficient, 1-min detection limit at 3σ of background signal during the laboratory characterization, and ambient background of selected amines and amides.

| Compounds | Proton affinity (kcal mol⁻¹) (NIST, 2016) | Sensitivity (mean ± σ) (Hz pptv⁻¹)[a] | Calibration coefficient ($10^{-2}$ MHz Hz⁻¹ pptv) | Detection limit (pptv) | Ambient background (mean ± σ) (pptv)[b] |
|---|---|---|---|---|---|
| Water | 165.2 | | | | |
| Ethanol | 185.6 | | | | |
| Ammonia | 204.0 | | | | |
| Methylamine (C$_1$-amine) | 214.9 | 7.06 ± 0.2 | 4.67 | 0.23 | 3.88 ± 1.23 |
| Dimethylamine (an isomer of C$_2$-amines) | 222.2 | 5.6 ± 0.2 | 5.89 | 0.50 | 6.64 ± 1.24 |
| Trimethylamine (an isomer of C$_3$-amines) | 226.8 | 19.4 ± 1.3 | 1.70 | 0.10 | 0.41 ± 0.14 |
| Diethylamine (an isomer of C$_4$-amines) | 227.6 | 6.4 ± 0.4 | 5.03 | 0.42 | 3.59 ± 1.04 |
| N,N-dimethyl-2-propanamine (an isomer of C$_5$-amines) | 232.0 | | | | 0.68 ± 0.32 |
| Triethylamine (an isomer of C$_6$-amines) | 234.7 | | | | 1.76 ± 0.79 |
| Formamide (an isomer of C$_1$-amide) | 196.5 | 38.0 ± 1.2 | 0.78 | 0.29 | 0.59 ± 0.50 |
| Acetamide (an isomer of C$_2$-amides) | 206.4 | 3.8 ± 0.3 | 7.89 | 0.45 | 8.63 ± 3.63 |
| N-methylformamide (an isomer of C$_2$-amides) | 203.5 | | | | |
| Propanamide (an isomer of C$_3$-amides) | 209.4 | 4.4 ± 0.1 | 6.82 | 1.95 | 59.76 ± 48.37 |
| N-methylacetamide (an isomer of C$_3$-amides) | 212.4 | | | | |
| N,N-dimethylformamide (an isomer of C$_3$-amides) | 212.1 | | | | |
| N-ethylacetamide (an isomer of C$_4$-amides) | 214.6 | | | | 13.59 ± 10.01 |
| N,N-dimethylacetamide (an isomer of C$_4$-amides) | 217.0 | | | | |
| 2,2-dimethyl-propanamide (an isomer of C$_5$-amides) | 212.5 | | | | 8.47 ± 5.18 |
| N,N-dimethylabutyramide (an isomer of C$_6$-amides) | 220.3 | | | | 2.60 ± 1.40 |

[a] Sensitivities were obtained under total reagent ion signals of ~ 0.32 MHz;
[b] Mean background values throughout the entire campaign ± one standard deviation for C$_1$-to C$_6$-amines and C$_1$-to C$_6$-amides.





Table 2 Inter-comparison of gaseous amines and amides measured in different locations with different surroundings.

| Location (site type, season) | $C_1$-Amine (pptv) | $C_2$-Amines (pptv) | $C_3$-Amines (pptv) | $C_4$-Amines (pptv) | $C_5$-Amines (pptv) | $C_6$-Amines (pptv) | $C_1$-Amide (pptv) | $C_2$-Amides (pptv) | $C_3$-Amides (pptv) | $C_4$-Amides (pptv) | $C_5$-Amides (pptv) | $C_6$-Amides (pptv) | Ref. |
|---|---|---|---|---|---|---|---|---|---|---|---|---|---|
| Hyytiälä, Finland (Forested, Spring) | | 12.2±7.7[a] | 59±35.5[a] | | | | | | | | | | Sellegri et al.(2005) |
| Hyytiälä, Finland (Forested, Summer and Autumn) | | 157±20[b] | 102±61[b] | 15.5±0.5[b] | | | | | | | | | Kieloaho et al.(2013) |
| Hyytiälä, Finland (Forested, Summer) | | 39.1[c] | 10.2[c] | 8.1[c] | | 1.6[c] | | | | | | | Hellén et al.(2014) |
| Alabama, USA (Forested, Summer) | <1.2 | <4.8 | 1-10 | <23.1 | <17.3 | <13.0 | | | | | | | You et al. (2014) |
| Kent, USA (Suburban, Winter) | <18 | 8±3[a] | 16±7[a] | <41 | | <8 | | | | | | | Yu and Lee (2012) |
| Kent, USA (Suburban, Summer) | 1-4 | <4.4 | 5-10 | 10-50 | 10-100 | <13.1 | | | | | | | You et al. (2014) |
| Southampton, UK (Suburban, Spring, Summer and Autumn) | | | | | | | | | 368-4357 | | | | Leach et al.(1999) |
| Lewes, USA (Coastal, Summer) | 5[c] | 28[c] | 6[c] | 3[c] | 1[c] | 2[c] | | | | | | | Freshour et al. (2014) |
| Lamont, USA (Continental, Spring) | 4[c] | 14[c] | 35[c] | 150[c] | 98[c] | 20[c] | | | | | | | Freshour et al. (2014) |
| Nanjing, China (Industrialized, Summer) | 0.1-18.9 | 0.1-29.9 | 0.1-9.3 | | | | | | | | | | Zheng et al. (2014) |
| Atlanta, USA (Urban, Summer) | <0.2 | 0.5-2 | 4-15 | ~5[d] | 4-5[d] | 3-25 | | | | | | | Hanson et al. (2011) |
| Helsinki, Finland (Urban, Summer) | | 23.6[c] | 8.4[c] | 0.3[c] | | 0.1[c] | | | | | | | Hellén et al. (2014) |
| Toroton, Canada (Urban, Summer) | | <2.7 | | <2.7 | | <1.0 | | | | | | | VandenBoer et al.(2011) |
| Shanghai, China (Urban, Summer) | 15.7±5.9[e] | 40.0±14.3[e] | 1.1±0.6[e] | 15.4±7.9[e] | 3.4±3.7[e] | 3.5±2.2[e] | 2.3±0.7[e] | 169.2±51.5[e] | 778.2±899.8[e] | 167.8±97.0[e] | 34.5±13.3[e] | 13.8±5.2[e] | This study |

[a] Mean values ± one standard deviation;
[b] The highest concentrations during the measurement;
[c] Mean values;
[d] 8 h average values;
[e] Mean values throughout the entire campaign ± one standard deviation.





**Figure captions:**



**Figure 1.** Influences of RH on the MS signals of methylamine (MA), trimethylamine (TMA), and
propanamide (PA).

**Figure 2.** Influences of organics on MS signals of methylamine (MA, panels A and B),
trimethylamine (TMA, panels A and B), and propanamide (PA, panels C and D). Note that the right
axis is used for the signal with an identical color, and other signals correspond to the left axis.

**Figure 3.** High-resolution single peak fitting for amines and amides.

**Figure 4.** Mass defect diagram for (A) protonated amines ($C_1$-$C_6$) and amides ($C_1$-$C_6$) and their
clusters with ethanol, together with other species with $m/z$ less than 163 Th in the ambient sample;
and (B) all nitrogen-containing species with $m/z$ less than 163 Th in the ambient sample. Circle
diameters are proportional to $\log_{10}$(count rates).

**Figure 5.** Time series of amines (panel A) and amides (panel B). Concentrations of amines and
amides are 15-min average values.

**Figure 6.** Time profiles of the rainfall, $C_2$-amines and $C_3$-amides.

**Figure 7.** The averaged diurnal profiles of $C_1$- and $C_2$-amines and $C_3$- and $C_4$-amides, together with
those of temperature, radiation, and ozone concentration during the campaign.

**Figure 8.** Three-day backward retroplumes (100 m above the ground level) from the sampling
location at (**A**) 05:00, 12 August 2015; (**B**) 21:00, 20 August 2015; (**C**) 06:00, 21 August 2015; and (**D**)
06:00, 25 August 2015. The embedded boxes show 12 h backward trajectories.


















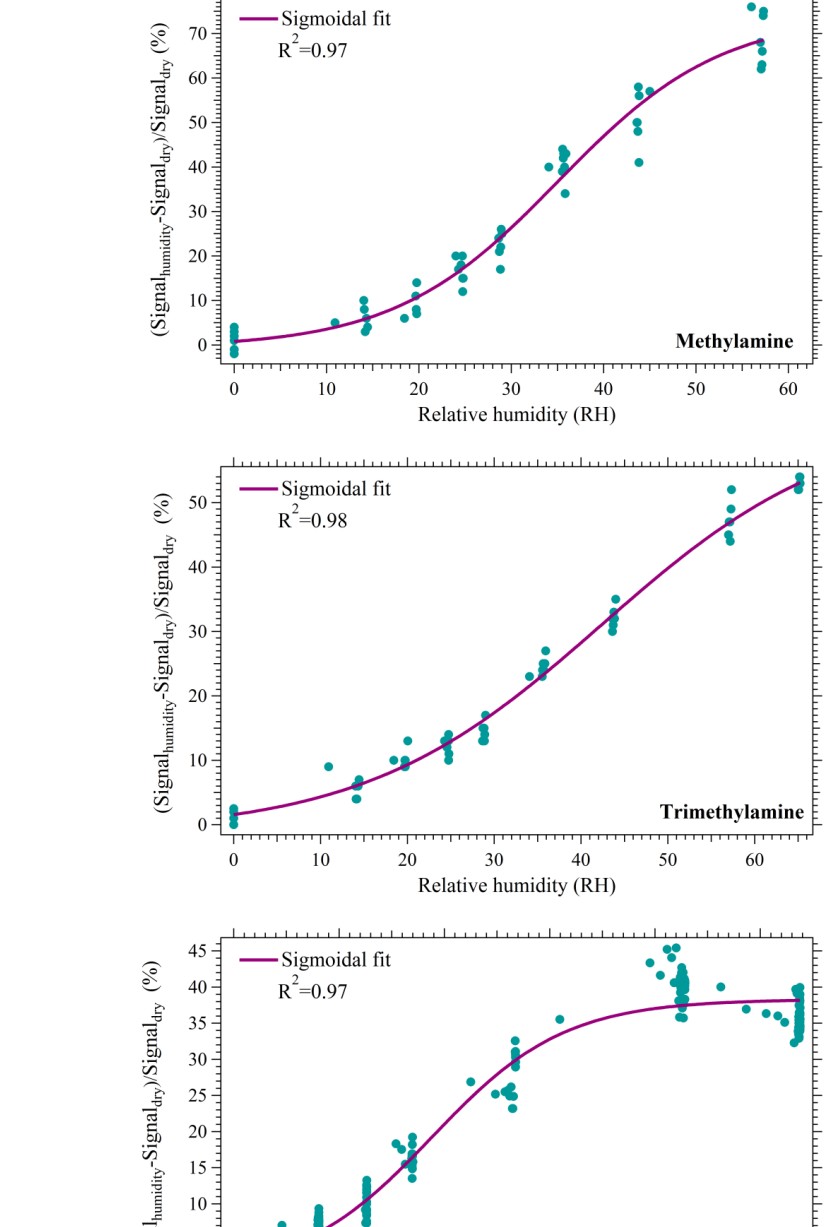





**Figure 1**





**Figure 2**






723        **Figure 3**




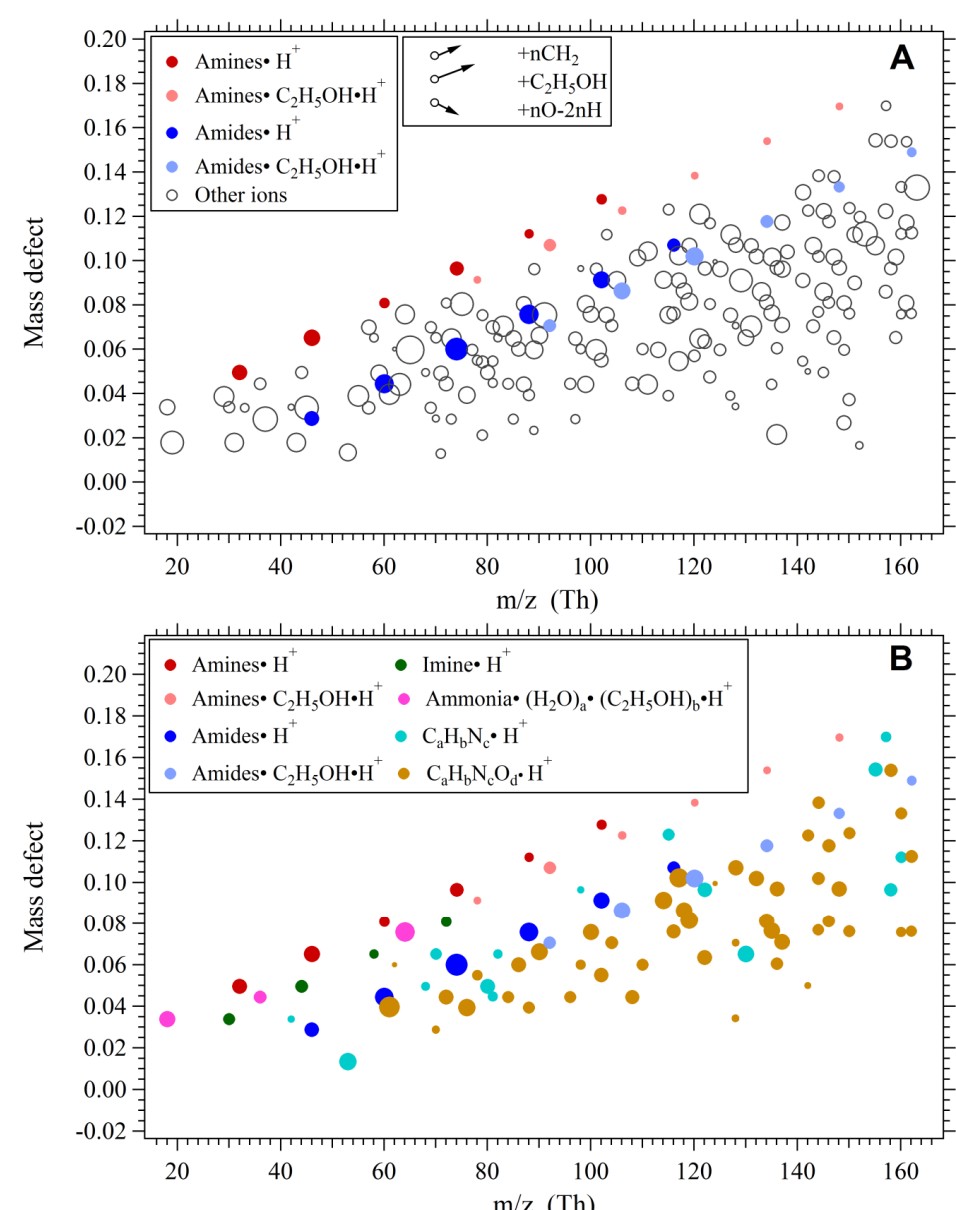


**Figure 4**



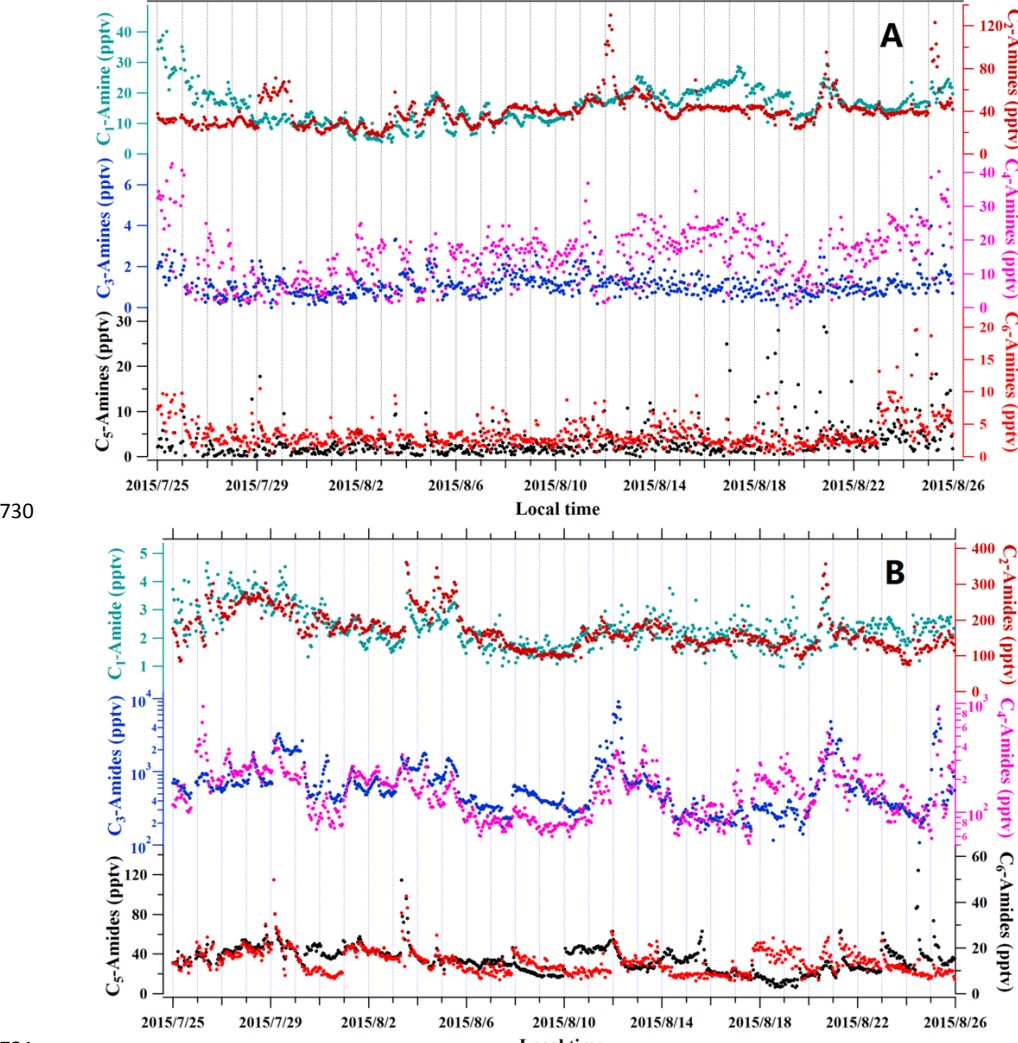



**Figure 5**






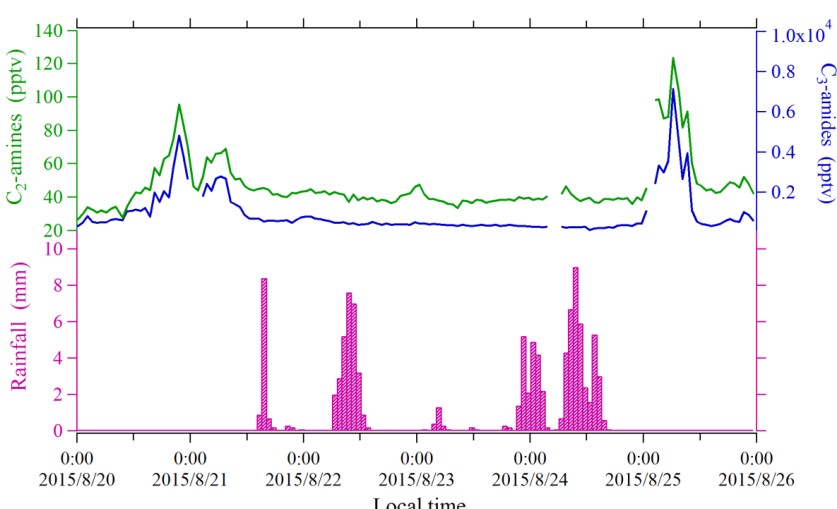


**Figure 6**

















**Figure 7**








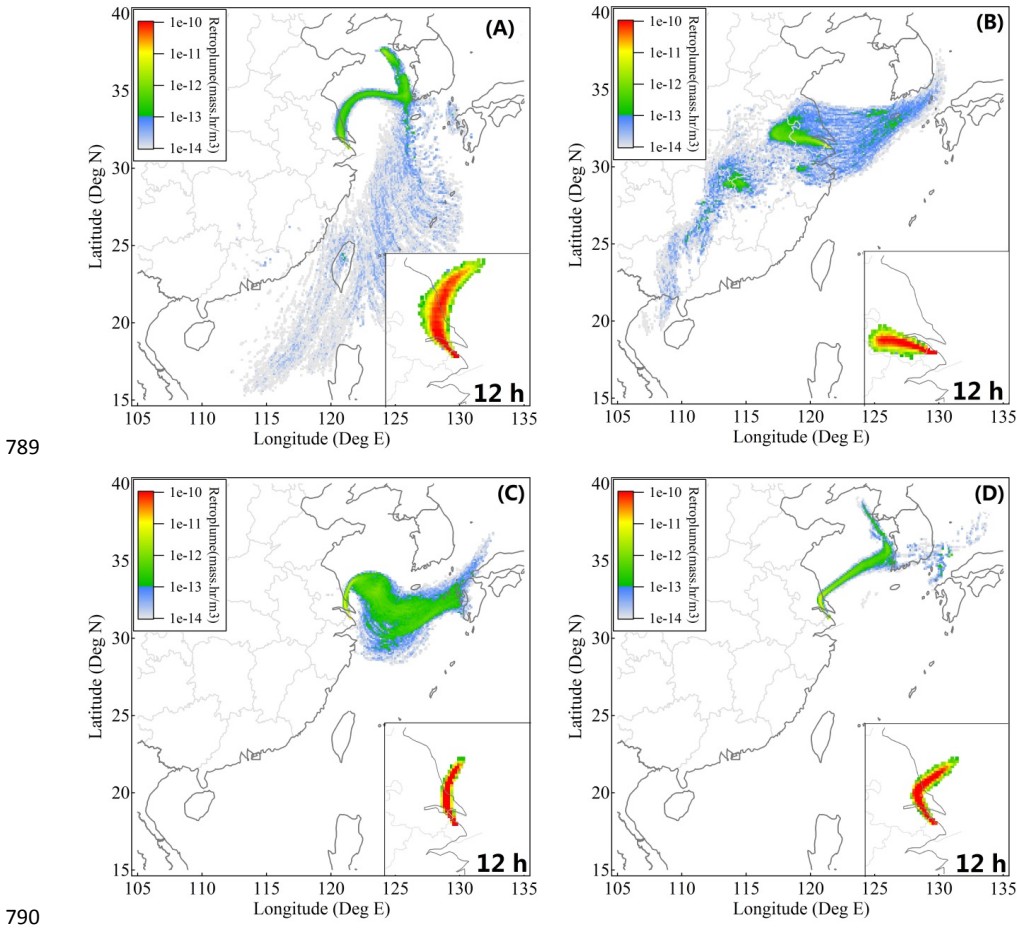


792        **Figure 8**