# Peer review of "Detection of atmospheric gaseous amines and amides by a high resolution time-of-flight chemical ionization mass spectrometer with protonated ethanol reagent ions"

_Atmospheric Chemistry and Physics, 2016_

## Referee Comment (RC1) · Anonymous Referee #1 · 8 Jul 2016

**Detection of atmospheric gaseous amines and amides by a high resolution time-of-flight chemical ionization mass spectrometer with protonated ethanol reagent ions**

**By Yao et al.**

The authors make simultaneous measurements of amines and amides using a recently developed online instrument, the HR-ToF-CIMS. They use ethanol as the reagent ion to optimize the detection of nitrogen-containing molecules. The instrument characteristics within the laboratory context are thorough and well presented, and include calibrations, relative humidity dependence measurements and organic effect on the measured concentrations.

The calibrated HR-ToF-CIMS was then employed to make ambient air measurements of a wide range of amines and amides at an urban site in Shanghai. The ambient measurements answer some questions about sources, sinks and transport of amines and amides and has the potential to be a useful reference for future work on the fate of organic nitrogen compounds in the atmosphere.

In particular, the manuscript is well presented with clear experimental detail. However, the referencing is generally incomplete and careful attention should be taken to research the literature accurately.

**General comments:**

- The introduction section describing previous measurements of amines and amides is a little tedious to read. Table 2 serves as a good summary and a clear reference and so perhaps the introduction could solely focus on identified trends in season, location, etc. It would also be important to keep referencing Table 2 in the discussion section. Finally, there could be value in making two tables, one for amines and one for amides. This arrangement would also highlight how few amide measurements exist. Also, see comment below for missing references.

- The referencing for previous work on amides is incomplete. Amides have been measured and speciated in the context of cigarette smoke and in charbroiling burgers. In addition, they have been identified in PM. See the following references for examples.

Schmeltz, I.; Hoffmann, D. Nitrogen-Containing Compounds in Tobacco and Tobacco Smoke. Chem. Rev. 1977, 77, 295−311

Rogge, W. F.; Hildemann, L. M.; Mazurek, M. A.; Cass, G. R.; Simoneit, B. R. T. Sources of Fine Organic Aerosol. 1. Charbroilers and Meat Cooking Operations. Environ. Sci. Technol. 1991, 25, 1112− 1125.

Sollinger, S.; Levsen, K.; Wünsch, G. Indoor Pollution by Organic Emissions from Textile Floor Coverings: Climate Test Chamber Studies Under Static Conditions. Atmos. Environ. 1994, 28, 2369− 2378.

Cheng, Y.; Li, S.; Leithead, A. Chemical Characteristics and Origins of Nitrogen-Containing Organic Compounds in PM2.5 Aerosols in the Lower Fraser Valley. Environ. Sci. Technol. 2006, 40, 5846−5852.

Laskin, A.; Smith, J. S.; Laskin, J. Molecular Characterization of Nitrogen-Containing Organic Compounds in Biomass Burning Aerosols Using High-Resolution Mass Spectrometry. Environ. Sci. Technol. 2009, 43, 3764−3771.

Raja, S.; Raghunathan, R.; Kommalapati, R. R.; Shen, X.; Collett, J. L., Jr.; Valsaraj, K. T. Organic Composition of Fogwater in the Texas−Louisiana Gulf Coast Corridor. Atmos. Environ. 2009, 43, 4214−4222.

- In Table S1, CHNO-H+ at m/z 44 is not reported and the authors explain that the proton affinity of HNCO is higher than that of ethanol in their response. Nonetheless, it would be important to mention this simply because of the growing interest in HNCO in our community.

  Roberts, J. M.; Veres, P. R.; Cochran, A. K.; Warneke, C.; Burling, I. R.; Yokelson, R. J.; Lerner, B.; Gilman, J. B.; Kuster, W. C.; Fall, R.; de Gouw, J. Isocyanic acid in the atmosphere and its possible link to smoke-related health effects. *Proc. Natl. Acad. Sci. U. S. A.* **2011***, 108*, 8966-8971.

- 2-aminoethanol (m/z 62) is an important industrial compound, especially in carbon capture and storage technologies and is listed in Table S1. Did the concentrations of this compound show high concentrations and/or wind speed/direction dependencies? Perhaps add a short discussion in section 3.2.1.

- The authors mention the use of the FIGAERO inlet and it is fair that they plan on reporting the particle measurements elsewhere. However, in the context of organic nitrogen it could be important to include a short discussion on how the gas phase and particle phase concentrations of organic nitrogen differ.

- How confident are the authors that their reported concentrations are solely gas phase? What possible contribution could there be from amines and amides in the particle phase?

- Many references and important discussions are included in the conclusion section but would perhaps be more appropriate in the discussion section.

**Specific comments**:

Lines 107-108: Could these authors specify what differences they are referring too? Diurnals? Absolute value? Etc.

Line 125: I believe that the Borduas et al. and Bunkan et al. references used PTR-MS to measure amides and amines and not CIMS. The authors should double check these references.

Line 170: Please clarify this sentence. "Dominantly protonated" compared to the water clusters?

Line 200: When the authors are referring to total ethanol signal, mean the sum of first three dimers with ethanol correct? (also, this sentence seems a bit out of place.)

Lines 296-304: Do the authors have a hypothesis as to why the enhancement is larger with amines than with amides? Perhaps because of differences in proton affinity?

Lines 305-312: What is the context of this experiment? Have others observed interferences with organics? Give context and appropriate references.

Lines 345-346: This is an important finding. Since a majority of nitrogen-containing compounds are oxygenated, it would be interesting to see examples of diurnals and to compare them with the amide diurnals. I would encourage the authors to emphasize this point and discuss further the implications of higher order nitrogen-containing compounds, including in the context of particle formation.

Line 363: "C5 and C6 amines standards are unavailable" is difficult to believe, especially since the authors build their own permeation tubes for the most part. I would recommend the authors clarify or remove this statement.

Line 383: a value of 778 +/- 899 is a little bizarre. How were the errors calculated?

Line 385-386: I would recommend moving this sentence to the paragraph discussing these high values in section 3.2.4.

Lines 393-397: What could be the role of particle phase chemistry on the observed diurnals?

Figure 2 – Can the authors estimate the concentrations of pinene and xylene used in these experiments? This information would be useful to include in the experimental section starting on line 210.

Figure 4 is an important analysis and can be used to discuss functional group interconversion and trends. I would recommend the authors spend a bit more time analyzing these graphs and discussing them.

Figure 6 is not convincing. The amines and amides seem to have decreased prior to rain fall. Are RH measurements available? Perhaps a better correlation can be seen with RH instead of rainfall?

Line 459: specify what is meant by "slightly better". Because of detection limits, or choice of reagent ion, or?

Lines 480-481: added references: Barnes et al., Borduas et al., Bunkan et al.

Missing reviews on organic nitrogen:

Cape, J. N.; Cornell, S. E.; Jickells, T. D.; Nemitz, E. Organic nitrogen in the atmosphere - Where does it come from? A review of sources and methods. *Atmos. Res.* **2011***, 102*, 30-48.

Fowler, D.; Coyle, M.; Skiba, U.; Sutton, M. A.; Cape, J. N.; Reis, S.; Sheppard, L. J.; Jenkins, A.; Grizzetti, B.; Galloway, J. N.; Vitousek, P.; Leach, A.; Bouwman, A. F.; Butterbach-Bahl, K.; Dentener, F.; Stevenson, D.; Amann, M.; Voss, M. The global nitrogen cycle in the twenty-first century. *Philos. Trans. R. Soc. London, Ser. B* **2013***, 368*.

Missing reference on imines:

Bunkan, A. J. C.; Tang, Y.; Selleveg, S. R.; Nielsen, C. J. Atmospheric gas phase chemistry of CH2=NH and HNC. A first-principles approach. *J. Phys. Chem. A* **2014***, 118*, 5279-5288.

One of a few more missing reference on gas-phase amine measurements:

Dawson, M. L.; Perraud, V.; Gomez, A.; Arquero, K. D.; Ezell, M. J.; Finlayson-Pitts, B. Measurement of gas-phase ammonia and amines in air by collection onto an ion exchange resin and analysis by ion chromatography. *Atmos. Meas. Tech.* **2014***, 7*, 2733-2744, 12 pp.

**Technical comments**:

Line 47: missing the word "the" before the word detection.

Lines 53-54: separate the sentences; one thought on the diurnal profile of amines and another on the diurnal profiles of amides.

Lines 88, 121: physio-chemical should be written as physi**co**-chemical. I think the authors are referring to physical properties and not to physiology properties.

Line 187: delete "referred to that as"

Lines 328-329: check syntax of the sentence.

Lines 433-436: check syntax of the sentence.

Line 487: progress is misspelled.

---

## Referee Comment (RC2) · Anonymous Referee #2 · 22 Aug 2016

Lines 93-94: C3 amines include trimethylamine, methyl-ethylamine and propylamine (formally also azetidine and the various methylaziridines, which have a different sum formula)

Lines 166-167: It is stated that the amines and amides reacted dominantly with protonated ethanol (through proton transfer reactions). This statement requires some clarification. The authors state in the lines preceeding that most abundant reagent ion is protonated ethanol dimer (illustrated in Fig. S1).

3.1.2 Effects of RH and organics: The influence of RH on the instrument sensitivity is

described and illustrated for three simple amines. The observed effect requires some explanation. Have the authors calculated the PTR rate coefficients for the different reagent ions? Does the reagent ion distribution change significantly ?

Line 314: "oxomide" should be "oxamide"

Figure 3 requires some additional information/clarification. First, the peak-shape is not discussed. Second, the curve-fitting includes a listing "Isotopes and other compounds". It is not obvious to the reader which "isotopes and other compounds" that are actually included in the analyses. As an example the region around m/z 60 must clearly contain a signal from the 13-C isotope of acetone (60.053), but this appears not to be the case.

––––––––––––––––––––––––––––

---

## Referee Comment (RC3) · Anonymous Referee #3 · 7 Sep 2016

**Review of Yao et al., Detection of atmospheric gaseous amines and amides by a high resolution time-of-flight chemical ionization mass spectrometer with protonated ethanol reagent ions**

The manuscript presents a method for the quantitative measurement of various amines and amides in the ppt range by Chemical Ionization using a High-Resolution Time-of-Flight mass spectrometer with protonated ethanol reagent ions. Calibrations are presented and the influence of humidity is characterized in the laboratory. Several weeks of ambient measurements in Shanghai are presented. 85 nitrogen-containing species with m/z < 163 Th are identified in the ambient air including numerous amines and amides. Amines reach mixing ratios up to more than 100 pptv. Amides reach maximum mixing ratios of several ppbv. Diurnal variations of specific amines and amides are studied.

The paper is well written, concise and well structured. Measurements are made with ethanol as reagent ions, a reagent ion that had not been tested in detail before. Atmospheric measurements of amides with CIMS have not been presented before. The paper is suitable for publication in ACP. Some minor comments should be taken into account.

**Minor comments:**

l. 86: some important earlier references are missing: Murphy et al., ACP, 7, 2313–2337, 2007; Kurten et al., ACP, 8, 4095–4103, 2008; Berndt et al., ACP, 10, 7101–7116, 2010.

l. 88: also Bzdek et al., ACP, 10, 3495–3503,-2010 and Kupianinen  et al., ACP, 12, 3591–3599, 2012, should be cited.

l. 103-105: compare also with the much lower amine concentrations measured with CIMS in Hyyttiälä as presented by Sipilä et al., AMT, 8, 4001–4011, 2015.

l. 122-123: Also Sipilä et al. 2015, and Simon et al., AMT, 9, 2135-2145, 2016, should be cited.

l 139-140. Avoid exact repetition of sentences from the abstract.

l. 366-367: compare also with the results of Sipilä et al., AMT, 2015.

Table 2: include also Sipilä et al., AMT, 8, 4001–4011, 2015, and Kürten et al., ACPD, doi:10.5194/acp-2016-294, 2016 in the inter-comparison.

Figure 3, upper right hand panel: some discussion of the "isotopes and other compounds" peaks needs to be given. Some more discussion of the uncertainties of the peak separation is necessary. Please discuss why the main peak needs to be separated into the two peaks as indicated. How large are the uncertainties in mass and signal intensity for the "isotopes and other compounds" peak?

**Technical corrections**

l. 5: omit comma between Yi-Jun and Liu

l. 158: …was THE protonated ethanol…

l. 159: … with the SECOND MOST dominant ions being THE protonated ethanol monomer…

l. 160: … and THE protonated ethanol trimer

l. 163: … the ratios of THE oxygen…

l. 168: (…$NR_3$, with R BEING EITHER A HYDROGEN ATOM or an alkyl group)

l. 169: (… WITH R` BEING EITHER A HYDROGEN or …)

l. 169-170: … can be REPRESENTED BY THE FOLLOWING REACTIONS (Yue…

l. 209: mixed WITH the amine/amide…

l. 359: …each DATA point…

l. 421: A Lagrangian…

l. 425: … are SHOWN for air masses

---

## Author Response (AR1)

**RE: A point-to-point response to Reviewer #1's comments**

"Detection of atmospheric gaseous amines and amides by a high resolution time-of-flight chemical ionization mass spectrometer with protonated ethanol reagent ions" (acp-2016-484) by Lei Yao, Ming-Yi Wang, Xin-Ke Wang, Yi-Jun Liu, Hang-Fei Chen, Jun Zheng, Wei Nie, Ai-Jun Ding, Fu-Hai Geng, Dong-Fang Wang, Jian-Min Chen, Douglas R. Worsnop, and Lin Wang

We are grateful to the helpful comments from this anonymous referee, and have carefully revised our manuscript accordingly. A point-to-point response to Reviewer #1's comments, which are repeated in italic, is given below.

**Reviewer #1's comments:**

*The authors make simultaneous measurements of amines and amides using a recently developed online instrument, the HR-ToF-CIMS. They use ethanol as the reagent ion to optimize the detection of nitrogen-containing molecules. The instrument characteristics within the laboratory context are thorough and well presented, and include calibrations, relative humidity dependence measurements and organic effect on the measured concentrations.*
*The calibrated HR-ToF-CIMS was then employed to make ambient air measurements of a wide range of amines and amides at an urban site in Shanghai. The ambient measurements answer some questions about sources, sinks and transport of amines and amides and has the potential to be a useful reference for future work on the fate of organic nitrogen compounds in the atmosphere.*
*In particular, the manuscript is well presented with clear experimental detail. However, the referencing is generally incomplete and careful attention should be taken to research the literature accurately.*

Reply: We are very grateful to the positive viewing of our manuscript by Reviewer #1, and have now revised our manuscript accordingly.

**General comments:**

1. *The introduction section describing previous measurements of amines and amides is a little tedious to read. Table 2 serves as a good summary and a clear reference and so perhaps the introduction could solely focus on identified trends in season, location, etc. It would also be important to keep referencing Table 2 in the discussion section. Finally, there could be value in making two tables, one for amines and one for amides. This arrangement would also highlight how few amide measurements exist. Also, see comment below for missing references.*

Reply: We have revised our manuscript accordingly. Table 2 has now been divided into two tables, and previous works on amides are shown in Table 3. The main text in the discussion has been revised accordingly.

2. *The referencing for previous work on amides is incomplete. Amides have been measured and speciated in the context of cigarette smoke and in charbroiling burgers. In addition, they have been identified in PM. See the following references for examples.*

Reply: We have added the related references, although particulate amides are not a focus of this study.

3. *In Table S1, CHNO-H+ at m/z 44 is not reported and the authors explain that the proton affinity of HNCO is higher than that of ethanol in their response. Nonetheless, it would be important to mention this simply because of the growing interest in HNCO in our community.*

Reply: Statements on detection of HNCO have been added in the main text, which reads (line 376-379),

 "One important atmospheric nitrogen-containing compound, isocyanic acid (HNCO) (Roberts et al., 2011) is not listed in Table S1, because the proton affinity of isocyanic acid is 180.0 kcal mol$^{-1}$, which is less than that of ethanol (185.6 kcal mol$^{-1}$). Hence, the ethanol reagent ions are not sensitive to the detection of isocyanic acid."

4. *2-aminoethanol (m/z 62) is an important industrial compound, especially in carbon capture and storage technologies and is listed in Table S1. Did the concentrations of this compound show high concentrations and/or wind speed/direction dependencies? Perhaps add a short discussion in section 3.2.1.*

Reply: No calibration was performed for 2- aminoethanol, which pre-excludes the determination of its concentration in this study. In addition, no correlation was found between 2-aminoethanol and C$_3$-amide.

5. *The authors mention the use of the FIGAERO inlet and it is fair that they plan on reporting the particle measurements elsewhere. However, in the context of organic nitrogen it could be important to include a short discussion on how the gas phase and particle phase concentrations of organic nitrogen differ.*

Reply: A pervious study reported that the concentrations of particulate primary amines measured with Fourier transform infrared spectroscopy (FTIR) from micron and submicron particles were 2 orders of magnitude higher than those of gas-phase amines (You et al., 2014). As a conclusion from that specific study, most of amines existed in particles.

We are now working on an optimized quantification method for particulate amines and amides, and other organic nitrogen compounds. It is premature from our dataset to make any statement on how the gas phase and particle phase concentrations of organic nitrogen differ at this stage.

6.  *How confident are the authors that their reported concentrations are solely gas phase? What possible contribution could there be from amines and amides in the particle phase?*

Reply: First of all, the FIGAERO inlet has two separate sampling ports: one for gases, and the other for particle collection and subsequent thermal desorption (Lopez-Hilfiker et al., 2014). The temperature of IMR (ion-molecule reactions chamber) was set at 50 °C to minimize the absorption and adsorption of amines and amides on the surface of IMR. Also, our inlet memory tests clearly suggest that the particle analysis (of amines and amides) will not have an impact on the gas sample analysis.

   We do not filter out particles in the flow when we analyze gas samples. However, given the pressure in IMR (~100 mbar) and the existence state of amines and amides in aerosols, particulate amines and amides are unlikely to evaporate and react with reagent ions (protonated ethanol ions) to make product ions that can be detected by the mass spectrometer. Hence, the interference for amines and amides from particles were negligible.

7.  *Many references and important discussions are included in the conclusion section but would perhaps be more appropriate in the discussion section.*

Reply: We have revised the conclusion section and emphasized on discussions in the discussion section. For example, we moved the following statement from the conclusion section to the discussion section (line 368-372).

   "Berndt et al. (2014) reported that pyridine was able to enhance nucleation in $H_2SO_4/H_2O$ system. Also, proton affinities of most of these heterocyclic nitrogen-containing compounds are higher than that of ammonia, hence they potentially have the capacity to neutralize atmospheric acidic species (e.g. $H_2SO_4$, $HNO_3$ and organic acids) to contribute to secondary particle formation and growth".

   Also, statements in the conclusion section have been moved into the introduction (line 89-91), "Compared with amines, acetamide has a very weak positive enhancement on the nucleation capability of sulfuric acid (Glasoe et al., 2015)."

**Specific comments:**

1.  *Lines 107-108: Could these authors specify what differences they are referring too? Diurnals? Absolute value? Etc.*

Reply: We meant to state that the absolute concentrations were not significantly different in different urban locations, which is clarified in our revised text

2.  *Line 125: I believe that the Borduas et al. and Bunkan et al. references used PTR-MS to measure amides and amines and not CIMS. The authors should double check these references.*

Reply: The principle of PTR-MS is similar to CIMS. Both of them utilize chemical ionization methodology to charge the analytes. Hence, PTR-MS is considered as a sub-category of CIMS.

3.  *Line 170: Please clarify this sentence. "Dominantly protonated" compared to the water clusters?*

Reply: We now state (Line 179-181) that "Amines and amides reacted dominantly with protonated ethanol ions $((C_2H_5OH)_n \cdot H^+$, n=1, 2 and 3), compared to water clusters, with the formation of the protonated amines/amides clustering with up to one molecule of ethanol".

4.  *Line 200: When the authors are referring to total ethanol signal, mean the sum of first three dimers with ethanol correct? (also, this sentence seems a bit out of place.)*

Reply: "Total ethanol signal" refers to the sum of the protonated ethanol monomer, dimer and trimer. The variation in total reagent ions between in laboratory calibration and during field measurements is taken into account when ambient concentrations of amines and amides are calculated. We clarify here (Line 211-214) that "The total ethanol reagent ion signals, i.e., the sum of the protonated ethanol monomer, dimer and trimer, during the laboratory calibration were typically ~0.32 MHz, which yields a small correction because of the variation in total reagent ions between in laboratory calibration and during field measurements as stated in section 2.4".

5.  *Lines 296-304: Do the authors have a hypothesis as to why the enhancement is larger with amines than with amides? Perhaps because of differences in proton affinity?*

Reply: We now state (Line 319-324) that "At elevated RH, high concentrations of water vapor result in the production of $(H_2O)_m H^+$ (m=1,2 and 3) and $C_2H_5OH \cdot H_2O \cdot H^+$ ions (Figure S1) (Nowak et al., 2002). Proton transfer reactions of $(H_2O)_m H^+$ (m=1,2 and 3) and $C_2H_5OH \cdot H_2O \cdot H^+$ with amines and amides might occur leading to the formation of additional protonated amines and amides. In addition, the proton affinities of amines are generally higher than those of amides. Thus, the relative enhancement was more significant for amines.

6.  *Lines 305-312: What is the context of this experiment? Have others observed interferences with organics? Give context and appropriate references.*

Reply: We now state (Line 325-328) that "Since a large number of ambient organics can be detected by this protonated ethanol reagent ion methodology as shown later, laboratory measurements of amines and amides in presence of and in absence of organics formed from photo oxidation of α-pinene and p-xylene, respectively, were carried out to examine the influence of organics on detection of amines and amides".

7.  *Lines 345-346: This is an important finding. Since a majority of nitrogen-containing compounds are oxygenated, it would be interesting to see examples of diurnals and to compare them with the amide diurnals. I would encourage the authors to emphasize this point and discuss further the implications of higher order nitrogen-containing compounds, including in the context of particle formation.*

Reply: This is an excellent suggestion, but determination of nitrogen-containing species with $m/z$ >163 from the ambient mass spectra is beyond the scope of this manuscript.

8. *Line 363: "C5 and C6 amines standards are unavailable" is difficult to believe, especially since the authors build their own permeation tubes for the most part. I would recommend the authors clarify or remove this statement.*

Reply: We now state (Line 396-397) that "Since the sensitivities of $C_5$- to $C_6$-amine were not determined, the sensitivity of DEA by HR-ToF-CIMS was adopted to quantify $C_5$- to $C_6$-amines"

9. *Line 383: a value of 778 +/- 899 is a little bizarre. How were the errors calculated? Line 385-386: I would recommend moving this sentence to the paragraph discussing these high values in section 3.2.4.*

Reply: The error here represents one standard deviation, which is derived from the significant fluctuation in the $C_3$-amide concentration as shown in Figure 5.

The sentence regarding the high concentration of $C_3$-amides has been moved to section 3.2.4.

10. *Lines 393-397: What could be the role of particle phase chemistry on the observed diurnals?*

Reply: According to our preliminary uptake experiments of amides by suspended $H_2SO_4$ particles and published uptake coefficient of amines by sulfuric acid (Wang et al., 2010), heterogeneous chemistry does not show a major preference between amines and amides. The lower RH in the daytime will generally lead to higher particle acidity, but not in the range of acidity that has been studied (Wang et al., 2010). Hence, the role of heterogeneous chemistry is not discussed here.

In the atmospheric condensed-phase material and cloud droplets, amines (e.g. methylamine) can react with aldehydes (e.g. methyglyoxal and glyoxal) to form oligomers that contribute to secondary organic aerosols (De Haan et al., 2009, 2011; Galloway et al., 2014). However, no data on aldehydes in the condensed-phase material and cloud droplets were available during our ambient campaign. Hence, we decided not to estimate the contribution of particle phase chemistry to the observed diurnals.

11. *Figure 2 – Can the authors estimate the concentrations of pinene and xylene used in these experiments? This information would be useful to include in the experimental section starting on line 210.*

Reply: The estimated concentrations of α-pinene and p-xylene were both ~200ppb. We have added the concentrations of these two compounds in main text.

12. *Figure 4 is an important analysis and can be used to discuss functional group interconversion and trends. I would recommend the authors spend a bit more time analyzing these graphs and discussing them.*

Reply: The high-resolution mass spectrometer allows us to determine the molecular formulas of these nitrogen-containing species and hence their mass defects, but would not permit a direct determination of the functional groups. The focus of this manuscript is detection of amines and amides. Interested readers can refer to the supplementary Table S1 that lists the tentative formula assignments for these nitrogen-containing species

13. *Figure 6 is not convincing. The amines and amides seem to have decreased prior to rain fall. Are RH measurements available? Perhaps a better correlation can be seen with RH instead of rainfall?*

Reply: In figure 6, the concentrations of amines and amides started to decrease right after the sunrise on August 21, 2015 (i.e., before the actual rain fall), which is in fact consist with their typical diurnal cycles. We meant to show in Figure 6 that the concentrations of amines and amides are generally maintaining at a low level on raining days, hinting the role of wet deposition. In addition, no clear correlation between RH and the concentrations of amines and amides were observed.

14. *Line 459: specify what is meant by "slightly better". Because of detection limits, or choice of reagent ion, or?*

Reply: We now state (Line 494-495) that "…are slightly better than those in previous studies using a similar protonated ethanol-CIMS method", which clarifies that both sensitivities and detection limits are slightly better. .

15. *Lines 480-481: added references: Barnes et al., Borduas et al., Bunkan et al.*

Reply: We have added these references.

**Technical comments:**

1. *Line 47: missing the word "the" before the word detection.*

Reply: We have revised our manuscript accordingly.

2. *Lines 53-54: separate the sentences; one thought on the diurnal profile of amines and another on the diurnal profiles of amides.*

Reply: We now state (Line 53-54) that "the diurnal profiles and backward trajectory analysis suggest that in addition to the secondary formation of amides in the atmosphere, industrial emissions could be important sources of amides in urban Shanghai".

3. *Lines 88, 121: physio-chemical should be written as physico-chemical. I think the authors are referring to physical properties and not to physiology properties.*

Reply: We have revised our manuscript accordingly.

*4. Line 187: delete "referred to that as" Lines 328-329: check syntax of the sentence.*

Reply: Line 187 (now Line 198): "referred to that as" has been removed.

Lines 328-329 (now lines 352-354): we now state that "In addition to the protonated $C_1$ to $C_6$-amines and amides, the presence of their clusters with one ethanol molecule is evident, which further confirms the identification of these species".

*5. Lines 433-436: check syntax of the sentence. Line 487: progress is misspelled.*

Reply: Lines 433-436 (now lines 468-471): we now state "Three-day backward retroplumes (100 m above the ground level) from the Fudan sampling site are shown for air masses with mixing ratios of $C_3$-amides > 2670 pptv in Figure 8, and for air masses with the $C_3$-amide concentration range between 1340 pptv and 2650 pptv in Figure S10, respectively."

Line 487: we revised our manuscript accordingly.

**References:**

Almeida, J., Schobesberger, S., Kürten, A., Ortega, I. K., Kupiainen-Määttä, O., Praplan, A. P., Adamov, A., Amorim, A., Bianchi, F., Breitenlechner, M., David, A., Dommen, J., Donahue, N. M., Downard, A., Dunne, E., Duplissy, J., Ehrhart, S., Flagan, R. C., Franchin, A., Guida, R., Hakala, J., Hansel, A., Heinritzi, M., Henschel, H., Jokinen, T., Junninen, H., Kajos, M., Kangasluoma, J., Keskinen, H., Kupc, A., Kurtén, T., Kvashin, A. N., Laaksonen, A., Lehtipalo, K., Leiminger, M., Leppä, J., Loukonen, V., Makhmutov, V., Mathot, S., McGrath, M. J., Nieminen, T., Olenius, T., Onnela, A., Petäjä, T., Riccobono, F., Riipinen, I., Rissanen, M., Rondo, L., Ruuskanen, T., Santos, F. D., Sarnela, N., Schallhart, S., Schnitzhofer, R., Seinfeld, J. H., Simon, M., Sipilä, M., Stozhkov, Y., Stratmann, F., Tome, A., Tröstl, J., Tsagkogeorgas, G., Vaattovaara, P., Viisanen, Y., Virtanen, A., Vrtala, A., Wagner, P. E., Weingartner, E., Wex, H., Williamson, C., Wimmer, D., Ye, P., Yli-Juuti, T., Carslaw, K. S., Kulmala, M., Curtius, J., Baltensperger, U., Worsnop, D. R., Vehkamäki, H., and Kirkby, J.: Molecular understanding of sulphuric acid-amine particle nucleation in the atmosphere, Nature, 502, 359-363, 10.1038/nature12663, 2013.

De Haan, D. O.; Tolbert, M. A.; Jimenez, J. L. Atmospheric condensed-phase reactions of glyoxal with methylamine. Geophys. Res. Lett. 36, L11819,2009.

De Haan, D. O.; Hawkins, L. N.; Kononenko, J. A.; Turley, J. J.; Corrigan, A. L.; Tolbert, M. A.; Jimenez, J. L. Formation of nitrogencontaining oligomers by methylglyoxal and amines in simulated evaporating cloud droplets. Environ. Sci. Technol. 45 (3), 984− 991,2011.

Galloway, M. M.; Powelson, M. H.; Sedehi, N.; Wood, S. E.; Millage, K. D.; Kononenko, J. A.; Rynaski, A. D.; De Haan, D. O. Secondary Organic Aerosol Formation during Evaporation of Droplets Containing Atmospheric Aldehydes, Amines, and Ammonium Sulfate. Environ. Sci.

Technol. 2014, 48 (24), 14417−14425.

Nowak, J. B.: Chemical ionization mass spectrometry technique for detection of dimethylsulfoxide and ammonia, J. Geophys. Res. Atmos., 107, 10.1029/2001jd001058, 2002.

Pratt, K. A., Hatch, L.E., Prather, K. A.: Seasonal volitility denpendence of ambient particle phase amines, Environ. Sci. Technol., 43, 5276-5281, 10.1021/es803189n, 2009.

Wang, L., Lal, V., Khalizov, A. F., and Zhang, R. Y.: Heterogeneous Chemistry of Alkylamines with Sulfuric Acid: Implications for Atmospheric Formation of Alkylaminium Sulfates, Environ. Sci. Technol., 44, 2461-2465, 10.1021/es9036868, 2010.

You, Y., Kanawade, V. P., de Gouw, J. A., Guenther, A. B., Madronich, S., Sierra-Hernández, M. R., Lawler, M., Smith, J. N., Takahama, S., Ruggeri, G., Koss, A., Olson, K., Baumann, K., Weber, R. J., Nenes, A., Guo, H., Edgerton, E. S., Porcelli, L., Brune, W. H., Goldstein, A. H., and Lee, S. H.: Atmospheric amines and ammonia measured with a chemical ionization mass spectrometer (CIMS), Atmos. Chem. Phys., 14, 12181-12194, 10.5194/acp-14-12181-2014, 2014.

**RE: A point-to-point response to Reviewer #2's comments**

"Detection of atmospheric gaseous amines and amides by a high resolution time-of-flight chemical ionization mass spectrometer with protonated ethanol reagent ions" (acp-2016-484) by Lei Yao, Ming-Yi Wang, Xin-Ke Wang, Yi-Jun, Liu, Hang-Fei Chen, Jun Zheng, Wei Nie, Ai-Jun Ding, Fu-Hai Geng, Dong-Fang Wang, Jian-Min Chen, Douglas R. Worsnop, and Lin Wang

We are grateful to the helpful comments from this anonymous referee, and have carefully revised our manuscript accordingly. A point-to-point response to Reviewer #2's comments, which are repeated in italic, is given below.

**Reviewer #2's comments:**

1. *Lines 93-94: C3 amines include trimethylamine, methyl-ethylamine and propylamine (formally also azetidine and the various methylaziridines, which have a different sum formula)*

Reply: The reported concentrations are the sum of propylamine (PA) and trimethylamine (TMA) (Kieloaho et al. 2013). This statement has been rephrased as "the sum of propylamine (PA) and trimethylamine (TMA)".

2. *Lines 166-167: It is stated that the amines and amides reacted dominantly with protonated ethanol (through proton transfer reactions). This statement requires some clarification. The authors state in the lines preceeding that most abundant reagent ion is protonated ethanol dimer (illustrated in Fig. S1).*

Reply: Under our CIMS setup, the most abundant reagent ion is the protonated ethanol dimer with the second most dominant ions being protonated ethanol monomer and trimer. The ion-molecule reactions between the analytes and reagent ions leads to the formation of protonated analytes (clustering with up to one molecule of ethanol), which is somehow similar to the conventional proton transfer reactions of hydronium ions. We now clarify (Line 179-181) that "Amines and amides reacted dominantly with protonated ethanol ions ($(C_2H_5OH)_n \cdot H^+$, n=1, 2 and 3), compared to water clusters, with the formation of protonated amines/amides clustering with up to one molecule of ethanol.".

3. *3.1.2 Effects of RH and organics: The influence of RH on the instrument sensitivity is described and illustrated for three simple amines. The observed effect requires some explanation. Have the authors calculated the PTR rate coefficients for the different reagent ions? Does the reagent ion distribution change significantly?*

Reply: As shown in Figure S1, $(H_2O)_m H^+$ (m=1, 2 and 3) and $C_2H_5OH \cdot H_2O \cdot H^+$ are present under elevated RH conditions, both of which are likely more reactive than protonated ethanol dimer (Nowak et al., 2002). Hence, proton transfer reactions of $(H_2O)_m H^+$ and $C_2H_5OH \cdot H_2O \cdot H^+$ with amines and amides may take place leading to the formation of protonated amines and amides. We have now (Line 319-322) state that

"At elevated RH, high concentrations of water resulted in the production of $(H_2O)_mH^+$ (m=1,2 and 3) and $C_2H_5OH \cdot H_2O \cdot H^+$ ions (Figure S1) (Nowak et al., 2002). Proton transfer reactions of $(H_2O)_mH^+$ (m=1, 2 and 3) and $C_2H_5OH \cdot H_2O \cdot H^+$ with amines and amides might occur leading to the formation of additional protonated amines and amides."

We didn't calculate the reaction rates of amines and amides with different reagent ions. Calibration curves with known concentrations of authentic standards were used to derive their ambient concentrations instead, although those curves were obtained at ~0% RH. However, in our ambient measurements, we used high purity nitrogen to dilute the sample flow resulting in an overall RH < 20%, under which RH no significant effect on reagent ions distribution was observed.

*4. Line 314: "oxomide" should be "oxamide"*

Reply: We have replaced "oxomide" by "oxamide".

*5. Figure 3 requires some additional information/clarification. First, the peak-shape is not discussed. Second, the curve-fitting includes a listing "Isotopes and other compounds". It is not obvious to the reader which "isotopes and other compounds" that are actually included in the analyses. As an example the region around m/z 60 must clearly contain a signal from the 13-C isotope of acetone (60.053), but this appears not to be the case.*

Reply: Clarification is now provided in the figure legend, the figure caption, and the main text. The exact formulas for "Isotopes and other compounds" have been provided in the figure legend. The new figure caption reads "**Figure 3.** High-resolution single peak fitting (custom shape) for amines and amides. During the peak deconvolution, only peaks whose areas are more than 0.5% of the total will be included in the figure legend."
Also, we state (Line 348-349) in the main text that "The assignment of molecular formulas for these species is within a mass tolerance of < 10 ppm, and the fitted area ranges from 99% to 101%".
We observed protonated acetone (m/z 59.0491) (Table S1) but its arbitrary signal was much less than that of protonated $C_2$-amide. Hence, the 13-C isotope of acetone, whose abundance is 3.3% of acetone, did not stand out in our mass spectra.

We are grateful to the helpful comments from this anonymous referee, and have carefully revised our manuscript accordingly. A point-to-point response to Reviewer #3's comments, which are repeated in italic, is given below.

**Reviewer #3's comments:**

*The manuscript presents a method for the quantitative measurement of various amines and amides in the ppt range by Chemical Ionization using a High‑Resolution Time‑of‑Flight mass spectrometer with protonated ethanol reagent ions. Calibrations are presented and the influence of humidity is characterized in the laboratory. Several weeks of ambient measurements in Shanghai are presented.85 nitrogen‑containing species with m/z < 163 Th are identified in the ambient air including numerous amines and amides. Amines reach mixing ratios up to more than 100 pptv. Amides reach maximum mixing ratios of several ppbv. Diurnal variations of specific amines and amides are studied.*
*The paper is well written, concise and well structured. Measurements are made with ethanol as reagent ions, a reagent ion that had not been tested in detail before. Atmospheric measurements of amides with CIMS have not been presented before. The paper is suitable for publication in ACP. Some minor comments should be taken into account.*

Reply: We are very grateful to the positive viewing of our manuscript by Reviewer #3, and have now revised our manuscript accordingly.

**Minor comments:**

1. *l. 86: some important earlier references are missing: Murphy et al., ACP, 7, 2313–2337, 2007; Kurten et al., ACP, 8, 4095–4103, 2008; Berndt et al., ACP, 10, 7101–7116, 2010.*

Reply: These three references have been added.

2. *l. 88: also Bzdek et al., ACP, 10, 3495–3503, 2010 and Kupianinen et al., ACP, 12, 3591–3599, 2012, should be cited.*

Reply: These two references have been added.

3. *l. 103‑105: compare also with the much lower amine concentrations measured with CIMS in Hyyttiälä as presented by Sipilä et al., AMT, 8, 4001–4011, 2015.*

Reply: Inter-comparison with results from Sipilä et al. (2015) is added. We now state (Line 114-117) that "Additionally, at the same site, DMA concentration was measured to be less than 150 ppqv (parts per quadrillion by volume) in May-June 2013 by an atmospheric pressure CIMS based on bisulfate-cluster method for DMA detection (Sipilä et al. 2015)."

4. *l. 122‐123: Also Sipilä et al. 2015, and Simon et al., AMT, 9, 2135‐2145, 2016, should be cited.*

Reply: These two references have been cited.

5. *l 139‐140. Avoid exact repetition of sentences from the abstract.*

Reply: We have revised our manuscript accordingly.

6. *l. 366‐367: compare also with the results of Sipilä et al., AMT, 2015.*

Reply: We have added the inter-comparison in discussion section 3.2.2 (line 407-411). We have now state that

"The concentrations of amines in Shanghai are generally smaller than those in Hyytiälä, Finland (Hellén et al., 2014; Kieloaho et al., 2013; Sellegri et al., 2005) except for one study that, as stated by the authors, should be treated with caution (Sipilä et al. 2015), potentially hinting that sources for amines existed in the forest region of Hyytiälä, Finland."

7. *Table 2: include also Sipilä et al., AMT, 8, 4001–4011, 2015, and Kürten et al., ACPD, doi:10.5194/acp‐2016‐294, 2016 in the inter‐comparison.*

Reply: These two references have been added. We have now state that (line 407-413)

"The concentrations of amines in Shanghai are generally smaller than those in Hyytiälä, Finland (Hellén et al., 2014; Kieloaho et al., 2013; Sellegri et al., 2005) except for one study that, as stated by the authors, should be treated with caution (Sipilä et al. 2015), potentially hinting that sources for amines existed in the forest region of Hyytiälä, Finland. Our $C_1$- and $C_2$-amines are generally more abundant than those in agricultural, coastal, continental, suburban, and urban areas (Freshour et al., 2014; Hanson et al., 2011; Kieloaho et al., 2013; Kürten et al., 2016; Sellegri et al., 2005; You et al., 2014)."

8. *Figure 3, upper right hand panel: some discussion of the "isotopes and other compounds" peaks needs to be given. Some more discussion of the uncertainties of the peak separation is necessary. Please discuss why the main peak needs to be separated into the two peaks as indicated. How large are the uncertainties in mass and signal intensity for the "isotopes and other compounds" peak?*

Reply: Clarification is now provided in the figure legend and the figure caption. The exact formulas for "Isotopes and other compounds" have been provided in the figure legend. The new figure caption reads "**Figure 3.** High-resolution single peak fitting (custom shape) for amines and amides. During the peak deconvolution, only peaks whose areas are more than 0.5% of the total will be included in the figure legend."

Also, we state (Line 348-349) in the main text that "The assignment of molecular formulas for these species is within a mass tolerance of < 10 ppm, and the fitted area ranges from 99% to 101%".

**Technical corrections:**

*1. l. 5: omit comma between Yi‑Jun and Liu*

Reply: We have revised our manuscript accordingly.

*2. l. 158: …was THE protonated ethanol…*

Reply: We have revised our manuscript accordingly.

*3. l. 159: … with the SECOND MOST dominant ions being THE protonated ethanol monomer…*

Reply: We have revised our manuscript accordingly.

*4. l. 160: … and THE protonated ethanol trimer*

Reply: We have revised our manuscript accordingly.

*5. l. 163: … the ratios of THE oxygen...*

Reply: We have revised our manuscript accordingly.

*6. l. 168: (…NR3, with R BEING EITHER A HYDROGEN ATOM or an alkyl group)*

Reply: We have revised our manuscript accordingly.

*7. l. 169: (… WITH R` BEING EITHER A HYDROGEN or …)*

Reply: We have revised our manuscript accordingly.

*8. l. 169‑170: … can be REPRESENTED BY THE FOLLOWING REACTIONS (Yue…*

Reply: We have revised our manuscript accordingly.

*9. l. 209: mixed WITH the amine/amide…*

Reply: We have revised our manuscript accordingly.

*10.   l. 359: …each DATA point…*

Reply: We have revised our manuscript accordingly.

*11.   l. 421: A Lagrangian…*

Reply: We have revised our manuscript accordingly.

*12.   l. 425: … are SHOWN for air masses*

Reply: We have revised our manuscript accordingly.

**Detection of atmospheric gaseous amines and amides by a high resolution**

**time-of-flight chemical ionization mass spectrometer with protonated**

**ethanol reagent ions**

[revised manuscript text omitted]

At elevated RH, high concentrations of water resulted in the production of $(H_2O)_mH^+$ (m=1,2 and

3) and $C_2H_5OH \cdot H_2O \cdot H^+$ ions (Figure S1) (Nowak et al., 2002). Proton transfer reactions of $(H_2O)_mH^+$

(m=1,2 and 3) and $C_2H_5OH \cdot H_2O \cdot H^+$ with amines and amides might occur leading to the formation of additional protonated amines and amides. In addition, the proton affinities of amines are generally higher than those of amides. Thus, the relative enhancement was more significant for amines.

Since a large number of ambient organics can be detected by this protonated ethanol reagent ion methodology as shown later, laboratory measurements of amines and amides in presence of and in absence of organics formed from photo oxidation of α-pinene and p-xylene, respectively, were carried out to examine the influence of organics on detection of amines and amides. Figure 2 shows the effects of biogenic (α-pinene) and anthropogenic (p-xylene) compounds and their photochemical reaction products on detection of amines (MA and TMA) and amide (PA) by our HR-ToF-CIMS.

[revised manuscript text omitted]

Cape, J. N., Cornell, S. E., Jickells, T. D., and Nemitz, E.: Organic nitrogen in the atmosphere — Where does it come from? A review of sources and methods, Atmos. Res., 102, 30-48, 10.1016/j.atmosres.2011.07.009, 2011.

Cheng, Y.; Li, S.; Leithead, A.:Chemical Characteristics and Origins of Nitrogen-Containing Organic Compounds in $PM_{2.5}$ Aerosols in the Lower Fraser Valley, Environ. Sci. Technol., 40, 5846−5852, 10.1021/es0603857, 2006,

Cornell, S. E., Jickells, T. D., Cape, J. N., Rowland, A. P., and Duce, R. A.:Organics nitrogen deposition on land
and coastal environments: a review od methods and data, Atmos. Environ., 37, 2173-2191,
10.1016/S1352-2310(03)00133-X, 2003.

Cox, R. A., and Yates, K.: The hydrolyses of benzamides, methylbenzimidatium ions, and lactams in aqueous
sulfuric-acid - the excess acidity method in the determination of reaction-mechanisms, Can. J. Chem., 59,
2853-2863, Doi 10.1139/V81-414, 1981.

Dawson, M. L., Perraud, V., Gomez, A., Arquero, K. D., Ezell, M. J., and Finlayson-Pitts, B. J.: Measurement of
gas-phase ammonia and amines in air by collection onto an ion exchange resin and analysis by ion
chromatography, Atmos. Meas. Tech., 7, 2733-2744, 10.5194/amt-7-2733-2014, 2014.

Draxler, R.R., and Hess,G.D.: An overview of the HYSPLIT_4 modeling system of trajectories, dispersion, and
deposition. Aust. Meteor. Mag., 47, 295-308,1998.

Ding, A. J., Wang, T., and Fu, C. B.: Transport characteristics and origins of carbon monoxide and ozone in
Hong Kong, South China, J. Geophys. Res. Atmos., 118, 9475-9488, 10.1002/jgrd.50714, 2013.

El Dib, G., and Chakir, A.: Temperature-dependence study of the gas-phase reactions of atmospheric NO3
radicals with a series of amides, Atmos. Environ., 41, 5887-5896, 10.1016/j.atmosenv.2007.03.038, 2007.

Erupe, M. E., Viggiano, A. A., and Lee, S. H.: The effect of trimethylamine on atmospheric nucleation involving
$H_2SO_4$, Atmos. Chem. Phys., 11, 4767-4775, 10.5194/acp-11-4767-2011, 2011.

Finlayson-Pitts, B. J., and Pitts, J. N.: Chemistry of the upper and lower atmosphere : theory, experiments, and
applications, Academic Press, San Diego, xxii, 969 p. pp., 2000.

Freshour, N. A., Carlson, K. K., Melka, Y. A., Hinz, S., Panta, B., and Hanson, D. R.: Amine permeation sources
characterized with acid neutralization and sensitivities of an amine mass spectrometer, Atmos. Meas. Tech.,
7, 3611-3621, 10.5194/amt-7-3611-2014, 2014.

Glasoe, W. A., Volz, K., Panta, B., Freshour, N., Bachman, R., Hanson, D. R., McMurry, P. H., and Jen.C.:
Sulfuric acid nucleation : an experimental study of the effect of seven bases, J. Geophys. Res. Atmos., 120,
1933-1950, 10.1002/2014JD022730, 2015.

Ge, X., Wexler, A. S., and Clegg, S. L.: Atmospheric amines – Part I. A review, Atmos. Environ., 45, 524-546,
10.1016/j.atmosenv.2010.10.012, 2011.

Hanson, D. R., McMurry, P. H., Jiang, J., Tanner, D., and Huey, L. G.: Ambient Pressure Proton Transfer Mass
Spectrometry: Detection of Amines and Ammonia, Environ. Sci. Technol., 45, 8881-8888,
10.1021/es201819a, 2011.

Hellén, H., Kieloaho, A. J., and Hakola, H.: Gas-phase alkyl amines in urban air; comparison with a boreal
forest site and importance for local atmospheric chemistry, Atmos. Environ., 94, 192-197,
10.1016/j.atmosenv.2014.05.029, 2014.

Kieloaho, A.-J., Hellén, H., Hakola, H., Manninen, H. E., Nieminen, T., Kulmala, M., and Pihlatie, M.:
Gas-phase alkylamines in a boreal Scots pine forest air, Atmos. Environ., 80, 369-377,
10.1016/j.atmosenv.2013.08.019, 2013.

Kim, H. A., Kim, K., Heo, Y., Lee, S. H., and Choi, H. C.: Biological monitoring of workers exposed to
N,N-dimethylformamide in synthetic leather manufacturing factories in Korea, Int. Arch. Occup. Environ.
Health, 77, 108-112, 10.1007/s00420-003-0474-1, 2004.

Kuhn, U., Sintermann, J., Spirig, C., Jocher, M., Ammann, C., and Neftel, A.: Basic biogenic aerosol precursors:
Agricultural source attribution of volatile amines revised, Geophys. Res. Lett., 38, L16811,
doi:10.1029/2011GL047958 , 2011.

Kupiainen, O., Ortega, I. K., Kurtén, T., and Vehkamäki, H.: Amine substitution into sulfuric acid – ammonia
clusters, Atmos. Chem. Phys., 12, 3591-3599, doi:10.5194/acp-12-3591-2012, 2012.

Kürten, A., Jokinen, T., Simon, M., Sipilä, M., Sarnela, N., Junninen, H., Adamov, A., Almeida, J., Amorim, A., Bianchi, F., Breitenlechner, M., Dommen, J., Donahue, N. M., Duplissy, J., Ehrhart, S., Flagan, R. C., Franchin, A., Hakala, J., Hansel, A., Heinritzi, M., Hutterli, M., Kangasluoma, J., Kirkby, J., Laaksonen, A., Lehtipalo, K., Leiminger, M., Makhmutov, V., Mathot, S., Onnela, A., Petäjä, T., Praplan, A. P., Riccobono, F., Rissanen, M. P., Rondo, L., Schobesberger, S., Seinfeld, J. H., Steiner, G., Tomé, A., Tröstl, J., Winkler, P. M., Williamson, C., Wimmer, D., Ye, P., Baltensperger, U., Carslaw, K. S., Kulmala, M., Worsnop, D. R., and Curtius, J.: Neutral molecular cluster formation of sulfuric acid-dimethylamine observed in real time under atmospheric conditions, Proc.Natl.Acad.Sci., 111, 15019-15024, 10.1073/pnas.1404853111, 2014.

Kürten, A., Bergen, A., Heinritzi, M., Leiminger, M., Lorenz, V., Piel, F., Simon, M., Sitals, R., Wagner, A., and Curtius, J.: Observation of new particle formation and measurement of sulfuric acid, ammonia, amines and highly oxidized molecules using nitrate CI-APi-TOF at a rural site in central Germany, Atmos. Chem. Phys. Discuss., doi:10.5194/acp-2016-294, in review, 2016.

Kurtén, T., Loukonen, V., Vehkamäki, H., and Kulmala, M.: Amines are likely to enhance neutral and ion-induced sulfuric acid-water nucleation in the atmosphere more effectively than ammonia, Atmos. Chem. Phys., 8, 4095-4103, doi:10.5194/acp-8-4095-2008, 2008.

Laskin A, Smith J S, Laskin J. Molecular characterization of nitrogen-containing organic compounds in biomass burning aerosols using high-resolution mass spectrometry, Environ. Sci. Technol., 43(10), 3764-3771, 10.1021/es803456n, 2009.

Leach, J., Blanch, A., and Bianchi, A. C.: Volatile organic compounds in an urban airborne environment adjacent to a municipal incinerator, waste collection centre and sewage treatment plant, Atmos. Environ., 33, 4309-4325, Doi 10.1016/S1352-2310(99)00115-6, 1999.

Lee, A., Goldstein, A. H., Kroll, J. H., Ng, N. L., Varutbangkul, V., Flagan, R. C., and Seinfeld, J. H.: Gas-phase products and secondary aerosol yields from the photooxidation of 16 different terpenes, J. Geophys. Res. Atmos., 111, Artn D1730510.1029/2006jd007050, 2006.

Lee, D., and Wexler, A. S.: Atmospheric amines – Part III: Photochemistry and toxicity, Atmos. Environ., 71, 95-103, 10.1016/j.atmosenv.2013.01.058, 2013.

Lopez-Hilfiker, F. D., Mohr, C., Ehn, M., Rubach, F., Kleist, E., Wildt, J., Mentel, T. F., Lutz, A., Hallquist, M., Worsnop, D., and Thornton, J. A.: A novel method for online analysis of gas and particle composition: description and evaluation of a Filter Inlet for Gases and AEROsols (FIGAERO), Atmos.Meas.Tech., 7, 983-1001, 10.5194/amt-7-983-2014, 2014.

Ma, Y., Xu, X., Song, W., Geng, F., and Wang, L.: Seasonal and diurnal variations of particulate organosulfates in urban Shanghai, China, Atmos. Environ., 85, 152-160, 10.1016/j.atmosenv.2013.12.017, 2014.

Malloy, Q. G. J., Qi, L., Warren, B., Cocker, D. R., Erupe, M. E., and Silva, P. J.: Secondary organic aerosol formation from primary aliphatic amines with $NO_3$ radical, Atmos. Chem. Phys., 9, 2051-2060, 2009.

Murphy, S. M., Sorooshian, A., Kroll, J. H., Ng, N. L., Chhabra, P., Tong, C., Surratt, J. D., Knipping, E., Flagan, R. C., and Seinfeld, J. H.: Secondary aerosol formation from atmospheric reactions of aliphatic amines, Atmos. Chem. Phys.,7, 2313-2337, doi:10.5194/acp-7-2313-2007, 2007.

Nielsen, C. J., Herrmann, H., and Weller, C.: Atmospheric chemistry and environmental impact of the use of amines in carbon capture and storage (CCS), Chem. Soc. Rev., 41, 6684-6704, 10.1039/c2cs35059a, 2012.

Nielsen, C. J., D'Anna, B., Karl, M., Aursnes, M., Boreave, A.,Bossi, R., Bunkan, A. J. C., Glasius, M., Hallquist, M., Hansen,A.-M. K., Kristensen, K., Mikoviny, T., Maguta, M. M., Müller,M., Nguyen, Q., Westerlund, J., Salo, K., Skov, H., Stenstrøm, Y.,and Wisthaler, A.: Atmospheric Degradation of Amines (ADA), Norwegian Institute for Air Research, Kjeller, Norway, 2011.

Nowak, J. B.: Chemical ionization mass spectrometry technique for detection of dimethylsulfoxide and ammonia, J. Geophys. Res. Atmos., 107, 10.1029/2001jd001058, 2002.

NIST: NIST Standard Reference Database Number 69, edited, National Institute for Standard Technology (NIST)

Chemistry Web Book, available at: http://webbook.nist.gov/chemistry/ (last access: 23 May 2016), 2016.

Qiu, C., Wang, L., Lal, V., Khalizov, A. F., and Zhang, R.: Heterogeneous reactions of alkylamines with ammonium sulfate and ammonium bisulfate, Environ. Sci. Technol., 45, 4748-4755, 10.1021/es1043112,

2011.

Roberts, J. M., Veres, P. R., Cochran, A. K., Warneke, C., Burling, I. R., Yokelson, R. J., Lerner, B., Gilman, J.

B., Kuster, W. C., Fall, R., and de Gouw, J.: Isocyanic acid in the atmosphere and its possible link to smoke-related health effects, Proc.Natl.Acad.Sci., 108, 8966-8971, 10.1073/pnas.1103352108, 2011.

Rogge, W. F., Hildemann, L. M., Mazurek, M. A., Cass, G. R., and Simonelt, B. R. T.: Sources Of Fine Organic

Aerosol .1. Charbroilers And Meat Cooking Operations, Environ. Sci. Technol., 25, 1112-1125, Doi

10.1021/Es00018a015, 1991.

Schmeltz, I., and Hoffmann, D.: Nitrogen-Containing Compounds In Tobacco And Tobacco-Smoke, Chem.Rev.,

77, 295-311, Doi 10.1021/Cr60307a001, 1977.

Sellegri, K., Hanke, M., Umann, B., Arnold, F., and Kulmala, M.: Measurements of organic gases during aerosol formation events in the boreal forest atmosphere during QUEST, Atmos. Chem. Phys., 5, 373-384, 2005.

Smith, D. F., Kleindienst, T. E., and McIver, C. D.: Primary product distributions from the reaction of OH with m-, p-xylene, 1,2,4- and1,3,5-trimethylbenzene,J.Atmos.Chem.,34,339-364,Doi10.1023/A:1006277328628,

1999.

Smith, J. N., Barsanti, K. C., Friedli, H. R., Ehn, M., Kulmala, M., Collins, D. R., Scheckman, J. H., Williams,

B. J., and McMurry, P. H.: Observations of aminium salts in atmospheric nanoparticles and possible climatic implications, Proc.Natl.Acad.Sci., 107, 6634-6639, 10.1073/pnas.0912127107, 2010.

Simon, M., Heinritzi, M., Herzog, S., Leiminger, M., Bianchi, F., Praplan, A., Dommen, J., Curtius, J., and

Kürten, A.: Detection of dimethylamine in the low pptv range using nitrate chemical ionization atmospheric pressure interface time-of-flight (CI-APi-TOF) mass spectrometry, Atmos. Meas. Tech., 9,

2135-2145, doi:10.5194/amt-9-2135-2016, 2016.

Sipilä, M., Sarnela, N., Jokinen, T., Junninen, H., Hakala, J., Rissanen, M. P., Praplan, A., Simon, M., Kürten, A.,

Bianchi, F., Dommen, J., Curtius, J., Petäjä, T., and Worsnop, D. R.: Bisulfate – cluster based atmospheric pressure chemical ionization mass spectrometer for high-sensitivity (< 100 ppqV) detection of atmospheric dimethyl amine: proof-of-concept and first ambient data from boreal forest, Atmos. Meas. Tech., 8,

4001-4011, doi:10.5194/amt-8-4001-2015, 2015.

[revised manuscript text omitted]
 (custom shape) for amines and amides. During the peak deconvolution, only peaks whose areas are more than 0.5% of the total will be included in the figure legend.

**Figure 4.** Mass defect diagram for (A) protonated amines ($C_1$-$C_6$) and amides ($C_1$-$C_6$) and their clusters with ethanol, together with other species with $m/z$ less than 163 Th in the ambient sample; and (B) all nitrogen-containing species with $m/z$ less than 163 Th in the ambient sample. Circle diameters are proportional to $\log_{10}$(count rates).

**Figure 5.** Time series of amines (panel A) and amides (panel B). Concentrations of amines and amides are 15-min average values.

**Figure 6.** Time profiles of the rainfall, $C_2$-amines and $C_3$-amides.

**Figure 7.** The averaged diurnal profiles of $C_1$- and $C_2$-amines and $C_3$- and $C_4$-amides, together with those of temperature, radiation, and ozone concentration during the campaign.

**Figure 8.** Three-day backward retroplumes (100 m above the ground level) from the sampling location at (**A**) 05:00, 12 August 2015; (**B**) 21:00, 20 August 2015; (**C**) 06:00, 21 August 2015; and (**D**) 06:00, 25 August 2015. The embedded boxes show 12 h backward trajectories.

[Figure]

[Figure]

[Figure]

**Figure 1**

[Figure]

**Figure 2**

[Figure]

[Figure]

[Figure]

**Figure 3**

**Figure 4**

[Figure]

[Figure]

[Figure]

**Figure 5**

[Figure]

                            **Figure 6**

[Figure]

**Figure 7**

[Figure]

**Figure 8**

Supplementary Materials:

**Detection of atmospheric gaseous amines and amides by a high resolution time-of-flight chemical ionization mass spectrometer with protonated ethanol reagent ions**

Lei Yao[1], Ming-Yi Wang[1,2], Xin-Ke Wang[1], Yi-Jun Liu[1,], Hang-Fei Chen[1], Jun Zheng[3], Wei Nie[5,], Ai-Jun Ding[4,], Fu-Hai Geng[6], Dong-Fang Wang[7], Jian-Min Chen[1], Douglas R. Worsnop[8], Lin Wang[1,]*

[1] *Shanghai Key Laboratory of Atmospheric Particle Pollution and Prevention (LAP[3]), Department of Environmental Science & Engineering, Fudan University, Shanghai 200433, China*

[2] *now at Center for Atmospheric Particle Studies, Carnegie Mellon University, Pittsburrgh, PA 15213, USA*

[3] *now at Pratt School of Engineering, Duke University, Durham, NC 27705,USA*

[4] *Jiangsu Key Laboratory of Atmospheric Environment Monitoring and Pollution Control, Nanjing University of Information Science & Technology, Nanjing 210044, China*

[5] *Joint International Research Laboratory of Atmospheric and Earth System Sciences, School of Atmospheric Science, Nanjing University, 210023, Nanjing, China*

[6] *Collaborative Innovation Center of Climate Change, Nanjing, Jiangsu Province, China*

[7] *Shanghai Meteorology Bureau, Shanghai 200135, China*

[8] *Shanghai Environmental Monitoring Center, Shanghai 200030, China*

[9] *Aerodyne Research, Billerica, MA 01821, USA*

*\* Corresponding Authors: L.W., email, lin_wang@fudan.edu.cn; phone, +86-21-65643568; fax, +86-21-65642080.*

**Table S1** Tentative formula assignment of MS peaks with *m/z* values less than 163 Th.

| ID | Ion formula | Molecular weight | Potential identity |
|---|---|---|---|
| 1 | $NH_3 \cdot H^+$ | 18.0338 | |
| 2 | $H_2O \cdot H^+$ | 19.0178 | |
| 3 | $HCN \cdot H^+$ | 28.0182 | Hydrogen cyanide |
| 4 | $C_2H_4 \cdot H^+$ | 29.0386 | |
| 5 | $CH_3N \cdot H^+$ | 30.0338 | |
| 6 | $CH_2O \cdot H^+$ | 31.0178 | |
| 7 | $O_2^+$ | 31.9893 | |
| 8 | $CH_5N \cdot H^+$ | 32.0495 | Methylamine |
| 9 | $CH_4O \cdot H^+$ | 33.0335 | |
| 10 | $NH_3 \cdot H_2O \cdot H^+$ | 36.0444 | |
| 11 | $(H_2O)_2 \cdot H^+$ | 37.0284 | |
| 12 | $C_2H_3N \cdot H^+$ | 42.0338 | Acetonitrile |
| 13 | $C_2H_2O \cdot H^+$ | 43.0178 | |
| 14 | $C_2H_5N \cdot H^+$ | 44.0495 | |
| 15 | $C_2H_4O \cdot H^+$ | 45.0335 | |
| 16 | $NO_2^+$ | 45.9924 | |
| 17 | $CH_3NO \cdot H^+$ | 46.0287 | Formamide |
| 18 | $C_2H_7N \cdot H^+$ | 46.0651 | $C_2$-Amine |
| 19 | $C_2N_2 \cdot H^+$ | 53.0134 | Cyanogen |
| 20 | $(H_2O)_3 \cdot H^+$ | 55.0389 | |
| 21 | $C_3H_4O \cdot H^+$ | 57.0335 | |
| 22 | $C_4H_8 \cdot H^+$ | 57.0699 | |
| 23 | $C_3H_7N \cdot H^+$ | 58.0651 | |
| 24 | $C_3H_6O \cdot H^+$ | 59.0491 | |
| 25 | $C_2H_3O_2 \cdot H^+$ | 60.0206 | |
| 26 | $C_2H_5NO \cdot H^+$ | 60.0444 | $C_2$-Amide |
| 27 | $C_3H_9N \cdot H^+$ | 60.0808 | $C_3$-Amine |
| 28 | $CH_4N_2O \cdot H^+$ | 61.0396 | |
| 29 | $C_2H_7NO \cdot H^+$ | 62.0600 | 2-Aminoethanol |
| 30 | $C_2H_6O_2 \cdot H^+$ | 63.0441 | |
| 31 | $(C_2H_5OH) \cdot NH_3 \cdot H^+$ | 64.0757 | |
| 32 | $(C_2H_5OH) \cdot H_2O \cdot H^+$ | 65.0597 | |
| 33 | $C_4H_5N \cdot H^+$ | 68.0495 | Pyrrole |
| 34 | $C_4H_4O \cdot H^+$ | 69.0335 | |
| 35 | $C_5H_8 \cdot H^+$ | 69.0699 | |
| 36 | $C_3H_3NO \cdot H^+$ | 70.0287 | |
| 37 | $C_4H_7N \cdot H^+$ | 70.0651 | 3-Pyrroline |
| 38 | $C_3H_2O_2 \cdot H^+$ | 71.0128 | |
| 39 | $C_4H_6O \cdot H^+$ | 71.0491 | |
| 40 | $C_3H_5NO \cdot H^+$ | 72.0444 | |
| 41 | $C_4H_9N \cdot H^+$ | 72.0808 | Pyrrolidine |
| 42 | $C_3H_4O_2 \cdot H^+$ | 73.0284 | |

| | | | |
|---|---|---|---|
| 43 | $C_4H_8O\cdot H^+$ | 73.0648 | |
| 44 | $C_3H_7NO\cdot H^+$ | 74.0600 | $C_3$-Amide |
| 45 | $C_4H_{11}N\cdot H^+$ | 74.0964 | $C_4$-Amine |
| 46 | $C_4H_{10}O\cdot H^+$ | 75.0804 | |
| 47 | $C_2H_5NO_2\cdot H^+$ | 76.0393 | Glycine |
| 48 | $C_3H_8O_2\cdot H^+$ | 77.0597 | |
| 49 | $C_2H_7NO_2\cdot H^+$ | 78.0549 | |
| 50 | $CH_3NH_2\cdot(C_2H_5OH)\cdot H^+$ | 78.0913 | |
| 51 | $C_2H_6OS\cdot H^+$ | 79.0212 | |
| 52 | $C_6H_6\cdot H^+$ | 79.0542 | |
| 53 | $C_3H_8O\cdot H_2O\cdot H^+$ | 79.0753 | |
| 54 | $C_5H_5N\cdot H^+$ | 80.0495 | Pyridine |
| 55 | $C_4H_4N_2\cdot H^+$ | 81.0447 | Pyrimidine |
| 56 | $C_2H_6O_2\cdot H_2O\cdot H^+$ | 81.0546 | |
| 57 | $C_6H_8\cdot H^+$ | 81.0699 | |
| 58 | $C_5H_7N\cdot H^+$ | 82.0651 | $N$-Methylpyrrole |
| 59 | $C_2H_6O\cdot(H_2O)_2\cdot H^+$ | 83.0703 | |
| 60 | $C_4H_5NO\cdot H^+$ | 84.0444 | |
| 61 | $C_4H_4O_2\cdot H^+$ | 85.0284 | |
| 62 | $C_5H_8O\cdot H^+$ | 85.0648 | |
| 63 | $C_4H_7NO\cdot H^+$ | 86.0600 | |
| 64 | $C_4H_6O_2\cdot H^+$ | 87.0441 | |
| 65 | $C_5H_{10}O\cdot H^+$ | 87.0804 | |
| 66 | $C_3H_5NO_2\cdot H^+$ | 88.0393 | $C_3$-Oxoamide |
| 67 | $C_4H_9NO\cdot H^+$ | 88.0757 | $C_4$-Amide |
| 68 | $C_5H_{13}N\cdot H^+$ | 88.1121 | $C_5$-Amine |
| 69 | $C_3H_4O_3\cdot H^+$ | 89.0233 | |
| 70 | $C_4H_8O_2\cdot H^+$ | 89.0597 | |
| 71 | $C_5H_{12}O\cdot H^+$ | 89.0961 | |
| 72 | $C_2H_7N_3O\cdot H^+$ | 90.0662 | |
| 73 | $C_4H_{10}O_2\cdot H^+$ | 91.0754 | |
| 74 | $CH_4NO\cdot(C_2H_5OH)\cdot H^+$ | 92.0706 | |
| 75 | $C_2H_7N\cdot(C_2H_5OH)\cdot H^+$ | 92.1070 | |
| 76 | $C_5H_5NO\cdot H^+$ | 96.0444 | |
| 77 | $C_5H_4O_2\cdot H^+$ | 97.0284 | |
| 78 | $C_6H_8O\cdot H^+$ | 97.0648 | |
| 79 | $C_5H_7NO\cdot H^+$ | 98.0600 | |
| 80 | $C_6H_{11}N\cdot H^+$ | 98.0964 | 2,5-Dimethyl-3-pyrroline |
| 81 | $C_5H_6O_2\cdot H^+$ | 99.0441 | |
| 82 | $C_6H_{10}O\cdot H^+$ | 99.0804 | |
| 83 | $C_5H_9NO\cdot H^+$ | 100.0757 | |
| 84 | $C_5H_8O_2\cdot H^+$ | 101.0597 | |
| 85 | $C_6H_{12}O\cdot H^+$ | 101.0961 | |
| 86 | $C_4H_7NO_2\cdot H^+$ | 102.0550 | $C_4$-Oxoamide |
| 87 | $C_5H_{11}NO\cdot H^+$ | 102.0913 | $C_5$-Amide |
| 88 | $C_6H_{15}N\cdot H^+$ | 102.1277 | $C_6$-Amine |
| 89 | $C_5H_{10}O_2\cdot H^+$ | 103.0754 | |
| 90 | $C_6H_{14}O\cdot H^+$ | 103.1117 | |

| | | | |
|---|---|---|---|
| 91 | $C_4H_9NO_2 \cdot H^+$ | 104.0706 | 4-Aminobutyric acid |
| 92 | $C_5H_{12}O_2 \cdot H^+$ | 105.0910 | |
| 93 | $C_2H_6NO \cdot (C_2H_5OH) \cdot H^+$ | 106.0863 | |
| 94 | $C_3H_9N \cdot (C_2H_5OH) \cdot H^+$ | 106.1226 | |
| 95 | $C_6H_5NO \cdot H^+$ | 108.0444 | |
| 96 | $C_8H_{12} \cdot H^+$ | 109.1012 | |
| 97 | $C_6H_7NO \cdot H^+$ | 110.0600 | |
| 98 | $C_6H_6O_2 \cdot H^+$ | 111.0441 | |
| 99 | $C_6H_8O_2 \cdot H^+$ | 113.0597 | |
| 100 | $C_6H_{11}NO \cdot H^+$ | 114.0913 | |
| 101 | $C_5H_6O_3 \cdot H^+$ | 115.0390 | |
| 102 | $C_6H_{10}O_2 \cdot H^+$ | 115.0754 | |
| 103 | $C_6H_{14}N_2 \cdot H^+$ | 115.1230 | 2,5-Dimethylpiperazine |
| 104 | $C_4H_5NOS \cdot H^+$ | 116.0165 | |
| 105 | $C_5H_9NO_2 \cdot H^+$ | 116.0760 | $C_5$-Oxoamide |
| 106 | $C_6H_{13}NO \cdot H^+$ | 116.1070 | $C_6$-Amide |
| 107 | $C_5H_8O_3 \cdot H^+$ | 117.0546 | |
| 108 | $C_6H_{12}O_2 \cdot H^+$ | 117.0910 | |
| 109 | $C_5H_{12}N_2O \cdot H^+$ | 117.1022 | |
| 110 | $C_5H_{11}NO_2 \cdot H^+$ | 118.0863 | L-Valine |
| 111 | $C_4H_{10}N_2O_2 \cdot H^+$ | 119.0815 | |
| 112 | $C_6H_{14}O_2 \cdot H^+$ | 119.1067 | |
| 113 | $C_5H_{11}OS \cdot H^+$ | 120.0570 | |
| 114 | $C_3H_7NO \cdot (C_2H_5OH) \cdot H^+$ | 120.1019 | |
| 115 | $C_4H_{12}N \cdot (C_2H_5OH) \cdot H^+$ | 120.1383 | |
| 116 | $C_8H_8O \cdot H^+$ | 121.0648 | |
| 117 | $C_6H_{14}O \cdot H_2O \cdot H^+$ | 121.1210 | |
| 118 | $C_4H_{11}NOS \cdot H^+$ | 122.0634 | |
| 119 | $C_8H_{11}N \cdot H^+$ | 122.0964 | Phenethylamine |
| 120 | $C_4H_{10}O_2S \cdot H^+$ | 123.0474 | |
| 121 | $C_8H_{10}O \cdot H^+$ | 123.0804 | |
| 122 | $C_9H_{14} \cdot H^+$ | 123.1168 | |
| 123 | $C_3H_9NO_2S \cdot H^+$ | 124.0995 | |
| 124 | $C_7H_8O_2 \cdot H^+$ | 125.0597 | |
| 125 | $C_8H_{12}O \cdot H^+$ | 125.0961 | |
| 126 | $C_6H_6O_3 \cdot H^+$ | 127.0390 | |
| 127 | $C_7H_{10}O_2 \cdot H^+$ | 127.0753 | |
| 128 | $C_8H_{14}O \cdot H^+$ | 127.1117 | |
| 129 | $C_5H_5NO_3 \cdot H^+$ | 128.0342 | |
| 130 | $C_6H_9NO_2 \cdot H^+$ | 128.0706 | |
| 131 | $C_7H_{13}NO \cdot H^+$ | 128.1070 | |
| 132 | $C_7H_{12}O_2 \cdot H^+$ | 129.0910 | |
| 133 | $C_9H_7N \cdot H^+$ | 130.0651 | Quinoline |
| 134 | $C_6H_{10}O_3 \cdot H^+$ | 131.0702 | |
| 135 | $C_7H_{14}O_2 \cdot H^+$ | 131.1067 | |
| 136 | $C_6H_{13}NO_2 \cdot H^+$ | 132.1019 | L-Leucine |
| 137 | $C_6H_{12}O_3 \cdot H^+$ | 133.0859 | |
| 138 | $C_3H_5NO_2 \cdot (C_2H_5OH) \cdot H^+$ | 134.0812 | |

| | | | |
|---|---|---|---|
| 139 | $C_4H_9NO\cdot(C_2H_5OH)\cdot H^+$ | 134.1176 | |
| 140 | $C_5H_{13}N\cdot(C_2H_5OH)\cdot H^+$ | 134.1539 | |
| 141 | $C_8H_6O_2\cdot H^+$ | 135.0441 | |
| 142 | $C_4H_{10}N_2O_3\cdot H^+$ | 135.0764 | |
| 143 | $C_6H_{14}O_3\cdot H^+$ | 135.1016 | |
| 144 | $C_7H_5NS\cdot H^+$ | 136.0215 | |
| 145 | $C_4H_9NO_4\cdot H^+$ | 136.0604 | |
| 146 | $C_5H_{13}NO_3\cdot H^+$ | 136.0968 | |
| 147 | $C_7H_8N_2O\cdot H^+$ | 137.0709 | |
| 148 | $C_9H_{12}O\cdot H^+$ | 137.0961 | |
| 149 | $C_6H_{14}O_2\cdot H_2O\cdot H^+$ | 137.1172 | |
| 150 | $C_4H_{11}NO_2S\cdot H^+$ | 138.1039 | |
| 151 | $C_7H_8O_3\cdot H^+$ | 141.0546 | |
| 152 | $C_8H_{12}O_2\cdot H^+$ | 141.0910 | |
| 153 | $C_9H_{16}O\cdot H^+$ | 141.1308 | |
| 154 | $C_6H_7NO_3\cdot H^+$ | 142.0499 | |
| 155 | $C_8H_{15}NO\cdot H^+$ | 142.1226 | |
| 156 | $C_7H_{10}O_3\cdot H^+$ | 143.0703 | |
| 157 | $C_8H_{14}O_2\cdot H^+$ | 143.1067 | |
| 158 | $C_8H_{18}N_2\cdot H^+$ | 143.1542 | 1-Butylpiperazine |
| 159 | $C_5H_9N_3O_2\cdot H^+$ | 144.0768 | |
| 160 | $C_7H_{13}NO_2\cdot H^+$ | 144.1019 | |
| 161 | $C_8H_{17}NO\cdot H^+$ | 144.1383 | |
| 162 | $C_6H_8O_4\cdot H^+$ | 145.0495 | |
| 163 | $C_7H_{12}O_3\cdot H^+$ | 145.0859 | |
| 164 | $C_8H_{16}O_2\cdot H^+$ | 145.1223 | |
| 165 | $C_6H_{11}NO_3\cdot H^+$ | 146.0811 | |
| 166 | $C_7H_{15}NO_2\cdot H^+$ | 146.1176 | 7-Aminoheptanoic acid |
| 167 | $C_6H_{10}O_4\cdot H^+$ | 147.0652 | |
| 168 | $C_7H_{14}O_3\cdot H^+$ | 147.1016 | |
| 169 | $C_8H_{18}O_2\cdot H^+$ | 147.1379 | |
| 170 | $C_4H_7NO_2\cdot(C_2H_5OH)\cdot H^+$ | 148.0968 | |
| 171 | $C_5H_{11}NO\cdot(C_2H_5OH)\cdot H^+$ | 148.1332 | |
| 172 | $C_6H_{15}N\cdot(C_2H_5OH)\cdot H^+$ | 148.1696 | |
| 173 | $C_5H_8O_3S\cdot H^+$ | 149.0267 | |
| 174 | $C_9H_8O_2\cdot H^+$ | 149.0597 | |
| 175 | $C_6H_{12}O_4\cdot H^+$ | 149.0808 | |
| 176 | $C_8H_7NS\cdot H^+$ | 150.0372 | |
| 177 | $C_5H_{11}NO_4\cdot H^+$ | 150.0761 | |
| 178 | $C_5H_{15}N_3O_2\cdot H^+$ | 150.1237 | |
| 179 | $C_5H_{14}N_2OS\cdot H^+$ | 151.0899 | |
| 180 | $C_{10}H_{14}O\cdot H^+$ | 151.1117 | |
| 181 | $C_7H_5NOS\cdot H^+$ | 152.0165 | |
| 182 | $C_{10}H_{15}O\cdot H^+$ | 152.1196 | |
| 183 | $C_6H_{14}O_3\cdot H_2O\cdot H^+$ | 153.1121 | |
| 184 | $C_9H_{14}O_2\cdot H^+$ | 155.1067 | |
| 185 | $C_9H_{18}N_2\cdot H^+$ | 155.1543 | 4-Pyrrolidinopiperidine |
| 186 | $C_8H_{12}O_3\cdot H^+$ | 157.0859 | |

| | | | |
|---|---|---|---|
| 187185 | $C_9H_{16}O_2 \cdot H^+$ | 157.1223 | |
| 188186 | $C_9H_{20}N_2 \cdot H^+$ | 157.1699 | Triacetonediamine |
| 189187 | $C_{11}H_{11}N \cdot H^+$ | 158.0964 | 2,8-Dimethylquinoline |
| 190188 | $C_9H_{19}NO \cdot H^+$ | 158.1539 | |
| 191189 | $C_7H_{10}O_4 \cdot H^+$ | 159.0652 | |
| 192190 | $C_8H_{14}O_3 \cdot H^+$ | 159.1016 | |
| 193191 | $C_{10}H_9NO \cdot H^+$ | 160.0757 | |
| 194192 | $C_{11}H_{13}N \cdot H^+$ | 160.1121 | 2,3,3-Trimethylindolenine |
| 195193 | $C_8H_{17}NO_2 \cdot H^+$ | 160.1332 | 8-Aminooctanoic acid |
| 196194 | $C_7H_{12}O_4 \cdot H^+$ | 161.0808 | |
| 197195 | $C_8H_{16}O_3 \cdot H^+$ | 161.1172 | |
| 198196 | $C_9H_{20}O_2 \cdot H^+$ | 161.1536 | |
| 199197 | $C_4H_5NO_3 \cdot (C_2H_5OH) \cdot H^+$ | 162.0761 | |
| 200198 | $C_5H_9NO_2 \cdot (C_2H_5OH) \cdot H^+$ | 162.1125 | |
| 201199 | $C_6H_{13}NO \cdot (C_2H_5OH) \cdot H^+$ | 162.1489 | |
| 202200 | $C_8H_{18}O_3 \cdot H^+$ | 163.1329 | |

[Figure]

Figure S1. A typical mass spectrum with RH < 20%. The dominant reagent ions are protonated ethanol dimer (($C_2H_5OH)_2 \cdot H^+$), monomer (($C_2H_5OH) \cdot H^+$), and trimer (($C_2H_5OH)_3 \cdot H^+$). The ratio of the clusters of protonated ethanol with water ($C_2H_5OH \cdot H_2O \cdot H^+$) to the sum of ($C_2H_5OH) \cdot H^+$, ($C_2H_5OH)_2 \cdot H^+$, and ($C_2H_5OH)_3 \cdot H^+$ is ~0.026.

[Figure]

[Figure]

Figure S2. Schematics for laboratory tests of effects of (**A**) RH and (**B**) organics. All sampling lines are made of PFA or PTFE material.

[Figure]

Figure S3. Schematic of CIMS setup during the field measurements.

[Figure]

Figure S4. Background check for amines and amides during a 3 h ambient sampling period.

[Figure]

Figure S5. Inlet memory of TMA and PA by inlet spike tests. The red and pink curves are fittings by the sum of two decaying exponentials. The characteristic decaying times of two exponentials, which are displacement of amines and amides inside the inlet by pumping and removing amines and amides adsorbed on the inlet surface, were 1.1 s and 8.5 s for TMA, and 1.4 s and 1.4 s for PA, respectively.

[Figure]

Figure S6. Changes in pH values of $HNO_3$ solution as titration by diethylamine proceeds. Note that pH values in this plot are 2-h averages

[Figure]

Figure S7. Changes in $NH_4^+$ concentration as hydrolysis of formamide and propanamide proceeds.

[Figure]

Figure S8. Calibration curves of $C_1$- to $C_4$-amines (MA, DMA, TMA, and DEA) and $C_1$- to $C_3$-amides (FA, AA, and PA). The red solid lines are linear fittings to guide the eye and the slopes are sensitivities for detection.

[Figure]

Figure S9. Diurnal variations of amines and amides with less variationss.

[Figure]

Figure S10. Three-day backward retroplumes (100 m above the ground level) from the sampling location at (**A**) 07:00, 27 July 2015; (**B**) 07:00, 28 July 2015; (**C**) 07:00, 30 July 2015; (**D**) 07:00, 31 July 2015; and (**E**) 07:00, 4 August 2015. The embedded boxes show 12h backward trajectories.